# A history of olive and grape cultivation in Southwest Asia using charcoal and seed remains

Katleen Deckers[1]*, Simone Riehl[2], Joseph Meadows[3], Valentina Tumolo[3,4], Israel Hinojosa-Baliño[3], Dan Lawrence[3]

1 Institute for Archaeological Sciences, University of Tübingen, Tübingen, Germany, 2 Institute for Archaeological Sciences and Senckenberg Center for Human Evolution and Palaeoenvironment (HEP), University of Tübingen, Tübingen, Germany, 3 Department of Archaeology, Durham University, Durham, United Kingdom, 4 Department of Humanistic Sciences, Communication and Tourism (DISUCOM), Università degli Studi della Tuscia, Viterbo, Italy

* katleen.deckers@uni-tuebingen.de

**Data Availability Statement:** All relevant data are within the manuscript and its Supporting information file.

## Abstract

Evaluating archaeobotanical data from over 3.9 million seeds and 124,300 charcoal fragments across 330 archaeological site phases in Southwest Asia, we reconstruct the history of olive and grape cultivation spanning a period of 6,000 years. Combining charcoal and seed data enables investigation into both the production and consumption of olive and grape. The earliest indication for olive and grape cultivation appears in the southern Levant around ca. 5000 BC and 4th millennium BC respectively, although cultivation may have been practiced prior to these dates. Olive and grape cultivation in Southwest Asia was regionally concentrated within the Levant until 600 BC, although there were periodic pushes to the East. Several indications for climate influencing the history of olive and grape cultivation were found, as well as a correlation between periods of high population density and high proportions of olive and grape remains in archaeological sites. While temporal uncertainty prevents a detailed understanding of the causal mechanisms behind these correlations, we suggest that long distance trade in olives, grapes and their associated products was integral to the economic, social, and demographic trajectories of the region.

## Introduction

It has been long argued that olive and grape cultivation played an important role in the development of past societies and has been further implicated in socio-economic processes such as the consolidation of power and the formation of elites, urbanization, imperial taxation, and trade [1–7]. However, no overview of the macrobotanical evidence for the region has been published, leaving many questions regarding connecting the domestication of olive and grape to their role in population changes and trade, and furthermore the extent of elites possibly driving these changes.

Genetic research locates a dominant olive domestication event in the Northern Levant near the Syrian/Turkish border [8] with multiple secondary events [9]; however, the earliest pollen

**Funding:** This research was supported by the European Research Council under the European Union's Horizon 2020 research and innovation program (https://research-and-innovation.ec. europa.eu/funding/funding-opportunities/funding-programmes-and-open-calls/horizon-2020_en) for the project 'CLaSS — Climate, Landscape, Settlement and Society: Exploring Human Environment Interaction in the Ancient Near East' (grant number 802424, award holder: DL). The work of VT was funded, from October 2022, by European Union funding under the Horizon Europe grant HORIZON-MSCA-2021-PF-01 (https://ec. europa.eu/info/funding-tenders/opportunities/ portal/screen/opportunities/topic-details/horizon-msca-2021-pf-01-01) for the project 'Food- Store' (grant agreement no. 101066771). The funders had no role in study design, data collection and analysis, decision to publish, or preparation of the manuscript.

**Competing interests:** The authors have declared that no competing interests exist.

evidence for olive cultivation from the Southern Levant (4500 BC) appears to predate the Northern Levant (2800 BC) [10]. Morphometric analysis of olive stones indicates weak domestication by the 5th millennium BC in the Carmel region, Israel [11], but likely by the 3rd millennium BC [12] several cultivated olive varieties existed and at the latest by the 13th century BC (Late Bronze Age) [13]. Recent DNA analysis for grape indicates two primary domestication centres ca 9000 BC, being Southwest Asia (Levant) and the Caucasus. However, archaeological finds do not support the early dates inferred from genomics [14] and pollen dynamics of cultivated vine are difficult to identify [15]. Shape analysis of grape seeds from the Middle Euphrates region shows many grape varieties in the 3rd millennium BC, amongst which a local variety of domestic grape, *Proles*, that was likely a hybrid of Asian and South Caucasian domestic vines [16].

Once olive and grape crops are established, return on investment is high. They provide an array of products that do not easily decay, like dried fruits, oil, and wine. Wine and olive oil especially have been intensively discussed as important trade products in ancient Southwest Asia which played a role in elite formation [3, 7, 17, 18]. The role of elites and the palace in the subsistence economy varied across this space and time and the extent to which elites/palaces were directly involved with fruit tree cultivation is often unclear [3, 4]. Another important issue relates to the economic interests of rulers regarding olive and grape. For example, scholars have argued that the westward expansion of the Assyrian Empire (858–627 BC) was related to economic interests in the production of olive oil, cf. [19] vs. [20]. For Judah however, regional differences were seen in high-gain high-risk specializations under the Assyrian rule, depending on the regional conditions, e.g., with the Shephelah specializing in olive, the highlands in viticulture, the Dead Sea valley on dates and exotic plants, while the Beersheba Valley on trade [21].

Besides the role of elites and empires in olive and grape production, there are also other anthropogenic factors that may have impacted the level of production of olive and grape, such as war and social unrest. For example, during the period of conflict at the end of the Late Bronze Age (ca. 1192–1175 BC), texts indicate that vineyards were destroyed at Ugarit in the Northern Levant [22]. Such instances of social unrest are thought to relate to climate change [23] which has been argued to have caused an overall system collapse at that time [22]. While some palynological studies have investigated the impact of climate change on fruit tree cultivation for specific regions and periods of Southwest Asia [24–27], no supra-regional investigations have been undertaken exploring the role of former rainfall and temperature in decisions related to investments in olive and grape cultivation and the possible consequences of climate change on crop yields and the nourishment of the population.

This paper aims to provide the first empirically supported diachronic overview of olive and grape cultivation in Southwest Asia covering the period from 6500 to 600 BC, a time frame well covered by the "Archaeobotanical Database of Eastern Mediterranean and Near Eastern Sites" (ADEMNES). The time sequence is characterized by important societal and population changes, such as the development of cities from the 4th millennium BC in some regions [28–30] and overall tendencies of increased settlement size, monumentality, and centralized administration of agricultural production [31–34]. These developments were halted ca. 2200 BC, when settlement disruptions are visible in many areas of Southwest Asia [28–30], possibly partially caused by aridity [35–37]. In the second millennium BC, city states again flourished, and territorial states started to form with ever changing political constellations [19]. Around 1200 BC ramifications within the population took place due to the collapse of major empires, for which climate has been argued to have played a role [22, 23], with various groups in the Levant adapting differently to the new circumstances [38]. While some regions saw no reestablishment of populations in the early Iron Age, others witnessed the emergence of Neo-Hittite states and the settling of pastoral nomadic groups [38]. Much of the Levant was fragmented while in the East

of Southwest Asia larger territorial states reemerged, such as the (Neo-) Assyrians. The latter rose to dominate the history of Southwest Asia [19] through incorporation of the Northern Levant by the 9[th] century BC and parts of the Southern Levant from the 8[th] century BC [38, 39] until it also collapsed around 609 BC, severe drought likely playing a role in this [40].

In this study we incorporate seed and charcoal data from the Southern and Northern Levant, Mesopotamia, and Iran to investigate the production and consumption of olive and grape. We use published and unpublished seed and charcoal data from archaeological sites in combination with geographical, palaeoclimatic and palaeodemographic data to generate new insights and interpretations. Integrating our archaeobotanical datasets with a model of the present-day natural distribution of wild olive and grape we locate possible areas that featured early cultivation. We also investigate the role of climate through time in the expansion and retraction of olive and grape cultivation and investigate its possible impact on the population. Additionally, we will gain insight into arboricultural practices, cultural differences, specialization, and the role of the palace in fruit tree cultivation. Finally, we will also consider trade of olive and grape products.

## Methods

To gain a regional and diachronic overview over cultivation of olive and grape, we use site by site archaeobotanical data from an updated version of the ADEMNES [41], alongside new unpublished charcoal results from sites including Kinet Höyük, Zincirli, Lachish, Hamoukar, and Jebel Mousa.

In total, archaeobotanical data from 328 site phases across 184 sites were evaluated, both as a whole as well as separated into regional groups (Northern Levant, Southern Levant, Mesopotamia, and Iran, see Fig 1). For some analyses we categorized sites into time blocks based on well-established sequences in each region, while for others we used the mid-point of the dated phase at the site. In total, data from more than 3,903,420 seed identifications and 124,300 charcoal identifications were assessed (Table 1).

For the seeds, all available data from ADEMNES was used, with some additional unpublished data (cf. S1 Table). Data was entered into ADEMNES as published. Site phases that were

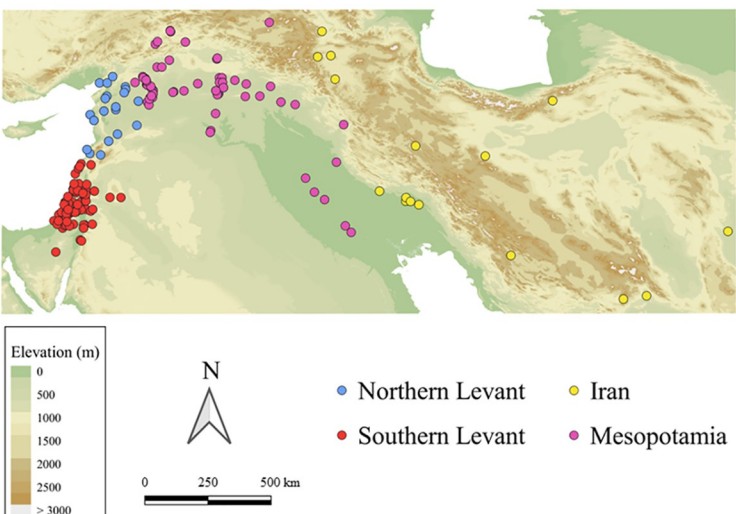

**Fig 1. Map of all archaeological sites whose archaeobotanical assemblages were evaluated in this manuscript grouped into their respective spatial definitions used in this manuscript.** Background SRTM DEM courtesy of the U.S. Geological Survey.

**Table 1. Seed and charcoal fragment counts, weights and volumes per time slice and region.** For a few sites charcoal numbers are approximations.

| Seed # | 6500–3600 BC | 3600–3000 BC | 3000–2500 BC | 2500–2000 BC | 2000–1500 BC | 1500–1200 BC | 1200–900 BC | 900–600 BC |
|---|---|---|---|---|---|---|---|---|
| Southern Levant | 94882 | 84455 | 184690 | 3389 | 144852 | 515868 | 1097940 | 269875 |
| Northern Levant | 74043 | 87 | 1846 | 55347 | 63605 | 13681 | 8066 | 7915 |
| Mesopotamia | 119678 | 27661 | 401633 | 116887 | 140827 | 31964 | 24203 | 360060 |
| Iran | 22214 | 82 | 6032 | 13200 | 446 | | 836 | 17159 |
| Seed fragment # sites | 6500–3600 BC | 3600–3000 BC | 3000–2500 BC | 2500–2000 BC | 2000–1500 BC | 1500–1200 BC | 1200–900 BC | 900–600 BC |
| Southern Levant | 16 | 24 | 11 | 2 | 20 | 25 | 29 | 19 |
| Northern Levant | 6 | 2 | 1 | 8 | 9 | 6 | 5 | 4 |
| Mesopotamia | 16 | 6 | 25 | 16 | 12 | 9 | 5 | 7 |
| Iran | 7 | 4 | 3 | 2 | 2 | | 1 | 7 |
| Charcoal fragment #, ml or gr | 6500–3600 BC | 3600–3000 BC | 3000–2500 BC | 2500–2000 BC | 2000–1500 BC | 1500–1200 BC | 1200–900 BC | 900–600 BC |
| Southern Levant | 1076 | 518 | 2518 | 166 | 2892 | 2743 | 2854/6022ml | 3628/2146 ml |
| Northern Levant | | 166 | 4384 | 708 | 10066 | 7559 | 1189 | 7833 |
| Mesopotamia | 4245 | 7052 | 7906 | 33085/1102gr | 1152/956gr | 9494/3327gr | 1317 | 15383 |
| Iran | 400 | 1620 | 2771 | 2019 | | | 28 | 978 |
| Charcoal fragment # sites | 6500–3600 BC | 3600–3000 BC | 3000–2500 BC | 2500–2000 BC | 2000–1500 BC | 1500–1200 BC | 1200–900 BC | 900–600 BC |
| Southern Levant | 4 | 7 | 4 | 1 | 7 | 7 | 7 | 9 |
| Northern Levant | | 1 | 1 | 2 | 6 | 3 | 1 | 2 |
| Mesopotamia | 6 | 3 | 9 | 8 | 2 | 4 | 3 | 4 |
| Iran | 1 | 1 | 2 | 1 | | | 1 | 3 |

represented by 2 or less samples were only included as presence/absence besides those that were published this way.

For the charcoals, site phases with less than 20 identifications were removed from the dataset. This is a low limit compared to the generally advisable minimal 250 fragments per site phase [42] but was employed here to retain many contemporary sites with low fragment counts in the Southern Levant which can be analyzed together to produce a general overview. Besides that, the cedar roof collapse samples from the palace of Tell Mishrifeh/Qatna were not included to the general charcoal dataset since they systematically do not represent local wood use and are rather exceptional compared to the other datasets that generally do not tend to represent large-scale samples of collapsed roofs.

While most of the archaeobotanical samples derive from domestic contexts, some were from non-domestic contexts, such as from palaces at Tell Mardikh/Ebla (2500–2000 BC), Tell Burak (2000–1500 BC), Atchana (1500–1200 BC), Tell Mishrifeh/Qatna (1500–1200 BC), a temple at Bazi (1500–1200 BC) and a not precisely identifiable monumental building at Zincirli (2000–1500 BC). These contexts are especially interesting for investigating the relationship between elites and production of *Vitis* and *Olea*. The contextual information of the plant remains allows us to reconstruct their final purpose (e.g., crop-processing by-products, storing of products, cooking, waste, fuel, etc.), while the find of grape and olive wood charcoal in palaces may be an indication for institutional engagement in production, especially if found in waste and fuel contexts.

Most archaeobotanical samples were collected through flotation, but some sites, especially in the Southern Levant, are represented by hand-picked samples. We included these, since mostly larger numbers of hand-picked samples were taken from many contexts. In charcoal studies it is known that handpicked samples often underrepresent minor taxa since larger fragments are selected, but still provide insights into the major taxa present [43].

For each site phase that was covered with seed data, percentages of *Olea* and *Vitis* seeds within crops were calculated as well as ubiquity percentages of the phase sample number. Since the late 1980s, most publications have used standardized methods for counting fragmented seeds. This involves adding two halves or four quarters of a seed as one count for most seeds. However, before this period, various counting methods were used, and some variable practices are still in use today for the counting of pistachio shells. These variable practices may introduce some inaccuracies to the quantification but still provide approximate insights. Seed proportion percentages provide some insight into the relative proportion of the crop at the site but can be misleading where a single context contains a very large number of seeds of one taxon, i.e., mainly stored products. Ubiquity percentages deliver additional information on the potential importance of *Olea* and *Vitis* since with a sufficiently large number and even distribution of samples over the excavation area these values reflect the overall percentage frequency of the taxon. Considering that grapes typically have 2–3 seeds while olive only one, seed percentages of *Olea* and *Vitis* may be not directly comparable, potentially causing an underrepresentation of olive if we think in terms of fruits. Grape seeds may also have been regularly swallowed by people and therefore may have followed different depositional histories than olive pips. Furthermore, their small size makes them less prone for being hand-picked by the excavators, which may counteract the above-mentioned higher number of seeds per fruit.

Canonical correspondence analysis (CCA) was applied on the crop plant dataset with chronology as the functional trait to detect diachronic changes using the program Canoco.

For the charcoals, the *Olea* and *Vitis* charcoal proportion percentage was calculated from the total charcoal fragment count of each site phase. In very few cases charcoals were published by weight rather than fragment count; these values were also used to calculate percentages and should produce comparable results [42]. According to the "Principle of least effort" paradigm it is generally assumed that wood -especially firewood- was collected in the direct surroundings of the site in proportion to its local environmental abundance [42]. *Olea* and *Vitis* charcoal proportions therefore can be used as proxies for pruning involvement in the surroundings of the site. Both provide a similar mass of pruning wood/cultivated ha today (0.6 ton/ha for grape vine; 0.5 to 1 ton/ha for olive [44].

Although absolute find numbers of archaeobotanical remains are a product of diverse pre- and post-depositional factors, we assume for this study that an increase in *Olea*/*Vitis* seed proportion and ubiquity indicates increased consumption or processing of the products of those plants by the inhabitants of a settlement, while an increase in *Olea*/*Vitis* charcoal may correlate with an increase in pruning waste and arboricultural involvement.

Large-scale and spatial patterns in the dataset can tell us about cultivation, trade, and exchange. Finds outside the wild ranges may indicate the expansion of arboriculture and could help in detecting earliest indications for cultivation, especially when there is evidence for charcoal and seeds, e.g. [45]. The presence of seeds but absence of charcoals, may be indicative of trade and exchange from regions where they were cultivated. We use the wild distributions for olive and grape as given in Zohary and Hopf [46]. Although recent research across several areas of our study region has demonstrated a wider distribution than Zohary and Hopf, especially for grape [16, 47–49], we apply the former to maintain regional consistency regarding the uneven distribution of recent studies, with the proviso that it likely represents an underestimation of the total distribution range.

Fruit tree proportions based on seed and charcoal finds are compared with reconstructed rainfall levels for the periods under consideration using the methodology described in Hewett et al. [50] (cf. S1 Table for reconstructed values for rainfall of the individual sites). Since olive requires at least 400 mm annual rainfall to grow well and produce reasonable harvests [26], we assume that those sites with olive evidence falling into precipitation ranges of less than 400

mm provide an indication of irrigation in case they were not imported. Aiming to elucidate possible vulnerability through human decision-making we also investigate whether changes can be observed in the distribution of sites with olive and grape in response to reconstructed rainfall levels, for example through expansion of fruit tree cultivation into regions with lower rainfall. We investigate whether sites with more rainfall show a higher degree of involvement with olive and grape. For this, linear regression was undertaken in combination with a variance analysis test (chosen significance level of 0.05).

Past population density is approximated here through normalized Summed Probability Distribution (SPD) of radiocarbon dates for the Levant and fitting additionally a logistic growth model to the observed SPD data with a 95% confidence envelope using Monte-Carlo methods (creating an MC envelope) as described in detail in [30]. Negative deviations from this MC envelope indicate periods of population decline greater than expected from the logistic growth model, while positive deviations indicate population growth greater than expected, suggestive of a growth beyond carrying capacity [30]. The R package *rcarbon* was used to process the radiocarbon data [51]. 2,468 radiocarbon dates were considered from 246 sites [52].

Besides rainfall, also temperature changes over time may have had an impact on the distribution and yields of olive and grape. This may be especially reflected in elevational changes of their distributions and proportions (cf. compare with [53]) since olive and grape are both cold-sensitive and cannot withstand strong frost [54]. Hence, upon warmer conditions, grape and olive cultivation may have expanded into higher elevations [55]. An additional aspect of interest regarding possible elevational changes for the grape and olive percentages is the presumed expansion of fruit tree cultivation into land considered marginal for other crops, due to their unevenness. We use elevation data for the sites (cf. S1 Table) from a basemap produced from SRTM 1 arc-second, courtesy of the U.S. Geological Survey.

To check for significant changes between the olive and grape records for the different periods and regions, the Kruskal-Wallis statistical test was undertaken with a follow-up Dunn's test. This was because unequal variances and not-normal distributions occurred. The Kruskal-Wallis test detects for significant differences for the median amongst independent groups based on ranks of the data. The Dunn's follow-up test then makes it possible to highlight the specific pairs of groups that show significant differences in median. Tests considered results statistically significant at a level of 0.05. Only datasets with 5 or more observations were used in the analysis (compare with Table 1). All statistical tests were performed in the program *jmp16*.

## Results

Fig 2. shows the presence/absence of olive seeds and/or charcoal according to region and period for those sites from which both the seeds and the charcoals were available. We assume sites with both olive charcoal and seeds, especially outside the natural distribution, indicate possibly places where local cultivation took place (Fig 2), while sites within the distribution of wild olive may also reflect the exploitation of wild olive. Moreover, sites where olive cultivation likely not took place had neither evidence of olive seeds nor wood charcoal (Fig 2) and those with seeds or charcoal may indicate import or local growth or both. The earliest indication for local cultivation of olive derives from the Southern Levant, from the sites Tell Tsaf and Tell esh-Shuna, and dates to ca. 5000 BC (i.e. 4750 and 4925 BC on Fig 2 that represents the middle of the occupation period) (cf. also [45]). At Nahal Zehora II, located within the present-day natural distribution, olive seeds and wood were found dating to the phase 6400–5200 BC (i.e. 5800 BC on Fig 2). First indication for local olive cultivation in the Northern Levant is later, only from the mid third millennium BC, and so far, no charcoals and seeds were investigated

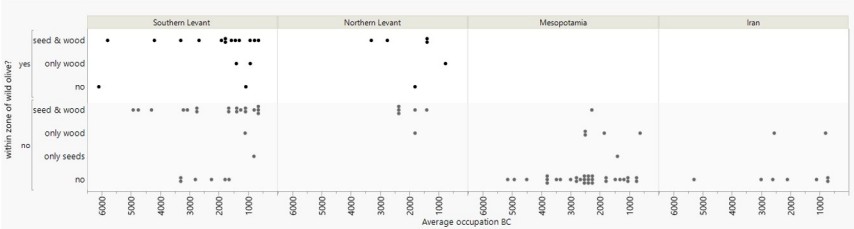

**Fig 2. Presence of olive charcoal and seeds in archaeological sites represented by dots divided into site locations within or outside the current zone of wild olives as defined by [46].**

from earlier sites outside the natural distribution. In Mesopotamia, both the charcoals and the seeds have been investigated from many early sites, but they show no evidence for olive cultivation. The earliest olive wood is present from the mid third millennium BC, while olive wood and olive stones are present at the site of Emar in the Euphrates Valley during the Middle Bronze Age (2000–1600 BC). In Iran, no strong evidence exists for local olive cultivation for the considered period. However, the site of Konar Sandal had olive wood, but no olive stones, and it has been argued it was another wild olive subspecies, namely *Olea europaea* subsp. *cuspidata* [56].

Fig 3 uses the same presence-absence model as Fig 2 but for grape. In the Southern Levant the earliest indication for grape cultivation is from around 3300–2900 BC from the site of Arad, while many sites from later periods have only grape seeds. In the Northern Levant, the earliest indication for grape cultivation outside the natural distribution dates to 2400–2200 BC, from Tell Mishrifeh/Qatna, but possible cultivation from within the wild distribution appears slightly earlier at Tell Fadous-Kfarabida around 3000–2500 BC. In Mesopotamia, the earliest grape vine seeds and charcoals are from Tell Brak, from the occupation phase at 4000–3600 BC. The earliest indication for grape cultivation in Iran is from Konar Sandal from the occupation phase 3100–2000 BC.

Figs 4 and 5 map all sites with *Olea*/*Vitis* seeds and/or charcoals, and therefore also include sites where wood or fruits may have been imported.

Regarding the *Olea* find distribution over time (Fig 4), even in the earliest phases there are several sites located close to but just outside the wild distribution mapped by [46]. From 3300 BC onwards, there is a rather different pattern (Fig 2) with an expansion further inland that lasts almost 2000 years (until 1200 BC). After 1400 BC, a retraction towards the Mediterranean coast is observable. To better visualize these observations on a Levant scale, see Fig 5.

For *Vitis* (Fig 6), several sites were found outside the Zohary and Hopf distribution [46], mostly along the major rivers. Of note is the absence of sites with *Vitis* in the Southern Levant

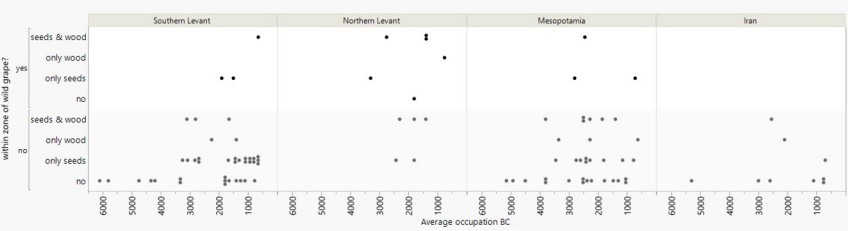

**Fig 3. Presence of grape charcoal and seeds in archaeological sites represented by dots, divided into site locations outside or within the current zone of wild grape as defined by [46].**

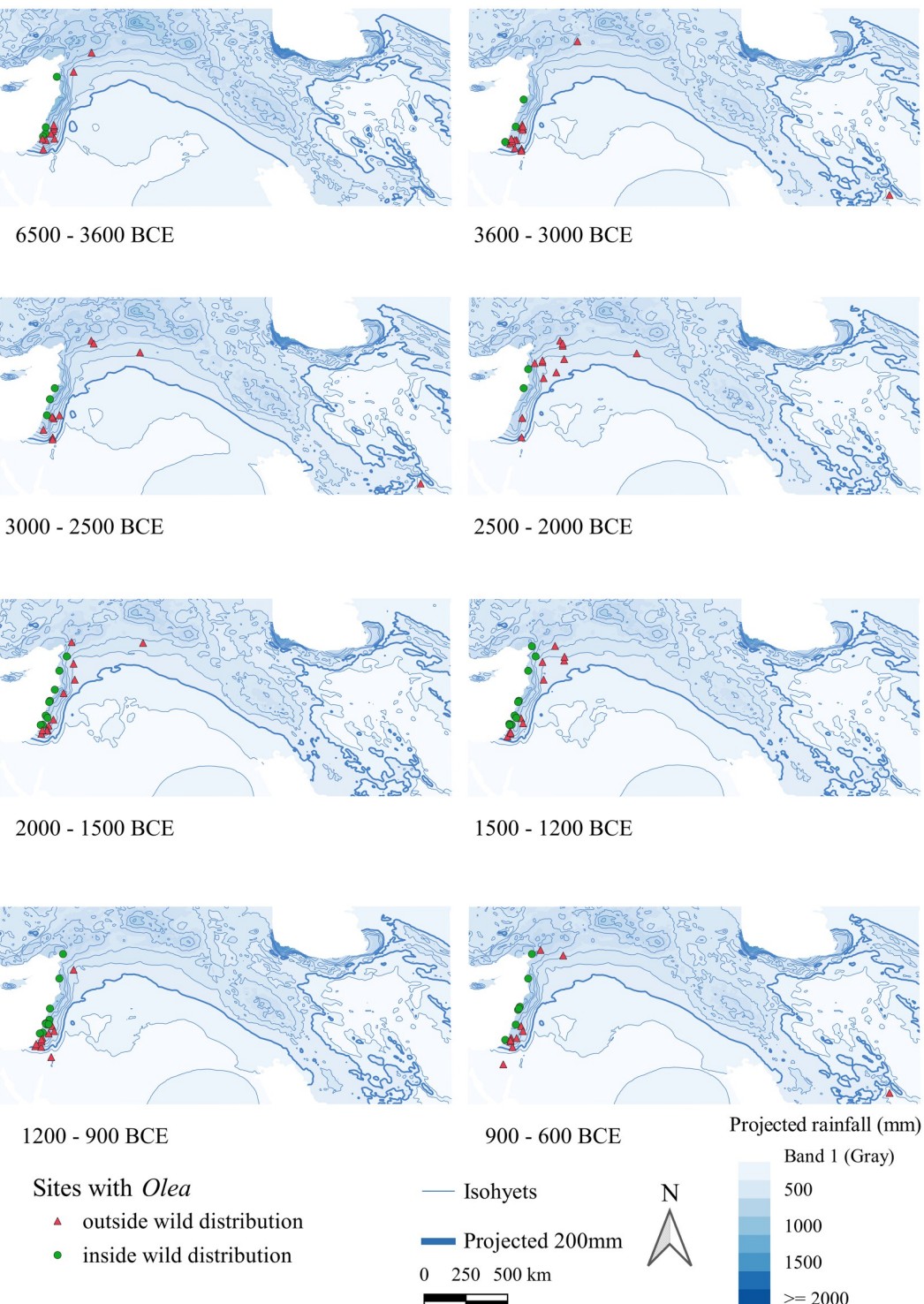

**Fig 4. Maps of archaeological sites with *Olea* finds, seeds or charcoal, indicating whether they fall within Zohary and Hopf's wild distribution [46].** The isohyets were calculated by the averaging of rainfall values hindcasted from Soreq Cave data within each respective period [50].

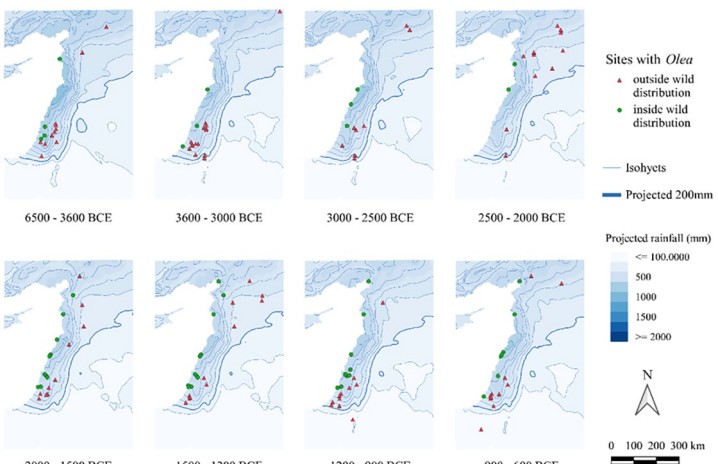

**Fig 5. Maps, with a particular focus on the Levant, of archaeological sites with *Olea* finds, seeds or charcoal, indicating whether they fall within Zohary and Hopf's wild distribution [46].** The isohyets were calculated by the averaging of rainfall values hindcasted from Soreq Cave data within each respective period [50].

between 6500 and 3600 BC, despite many sites investigated in this region. From 3600–3000 BC, however, most of the sites are located in the Southern Levant and outside the mapped wild distribution range. In the phase 3000–2500 BC, there appear more sites with *Vitis* in northern Mesopotamia with a wide distribution regarding rainfall, even reaching south of the 200 mm isohyet along the Euphrates. In the next phase, sites with *Vitis* contract to areas with higher rainfall levels. Within the phases 1200–900 BC and 900–600 BC, we again see southwards incursions of grape cultivation along the major rivers. To better visualize these Levantine trends, see Fig 7. In Iran, the site of Tepe Yahya has evidence for *Vitis* seeds as early as the phase 4500–3800 BC and continued to have some in subsequent phases. From the phase 900–600 BC there is a strong representation of sites with *Vitis* in Iran.

Fig 8 shows proportional changes over time in percentages of *Olea* seeds among crop plants, *Olea* seed ubiquities and *Olea* charcoal among all charcoals at sites. These are proxies for the intensity of involvement with olive. The Northern and Southern Levant show strong involvement with olive that appears different from Mesopotamia and Iran. A Kruskal-Wallis test shows statistically significant differences in the distributions between periods of individual regions amongst the groups (Chi square for olive % of crop = 77.7355, p = < .0001, df = 21; Chi square for olive seed ubiquity % = 69.7350, p = <0.001, df = 21), with the follow-up Dunn's test showing significant differences as depicted in Table 2. The major outcome from the combination of Table 2 and Fig 8 is that the Southern Levant and Mesopotamia differ regarding olive. The absence of the detection of more statistically significant differences, especially for phases within a region, does not imply that there are no differences; rather, it likely relates to the rather low sample sizes and less extreme differences, reducing the power of the test.

For the olive charcoal, significant differences were detected on the dataset through Kruskal-Wallis analysis (Chi square = 37.0291, p = < .0001, df = 11), while the Dunn's test showed no significant differences. Though, overall, the charcoal dataset covers much less site phases per region and period than the seed dataset (Table 1), possibly causing an inability to detect significant changes.

Fig 9 shows boxplots for the *Vitis* seed and charcoal percentages at the sites per phase and region. A Kruskal-Wallis test was done to investigate possible significant differences between

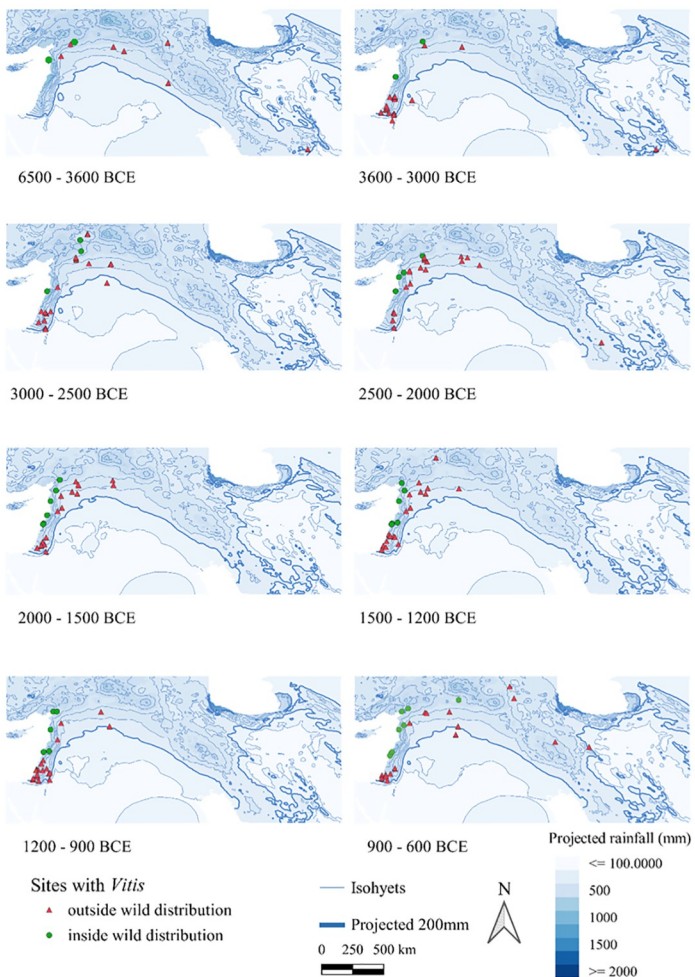

**Fig 6. Maps of archaeological sites with *Vitis* finds, seeds or charcoal, with an indication as to whether within the wild distribution as mapped by in Zohary and Hopf [46].** The isohyets are based on average values for the period under consideration obtained by the hindcasting of Soreq Cave data [50].

phases and regions. The test indicated significant differences at the level 0.05 in the distribution of the samples (for grape seed % of crops Chi square = 59.5052, p = <0.001, df = 21; for grape seed ubiquity % Chi square = 56.1467, p<0.0001, df = 20). A follow-up Dunn's test indicated significant differences between some regions and phases (Table 2). Here again, the rather low sample number gives the test reduced statistical power to prove visually observable differences. However, of note is that the test could detect several times that the Southern Levant in the phase 6500–3600 BC stands out as significantly different from later periods in the same region and the Northern Levant, underlining the observed Southern Levantine absence of grape in this period as significantly different from later.

For the charcoals, *Vitis* percentages were overall much lower than *Olea* percentages. A Kruskal-Wallis test could not detect significant differences at the level 0.05 in the distribution of the samples. It is possible that the lack of difference between the phases is real, but it should be noted that the overall site phases covered by the charcoals are much lower than for the seeds (cfr. Table 1).

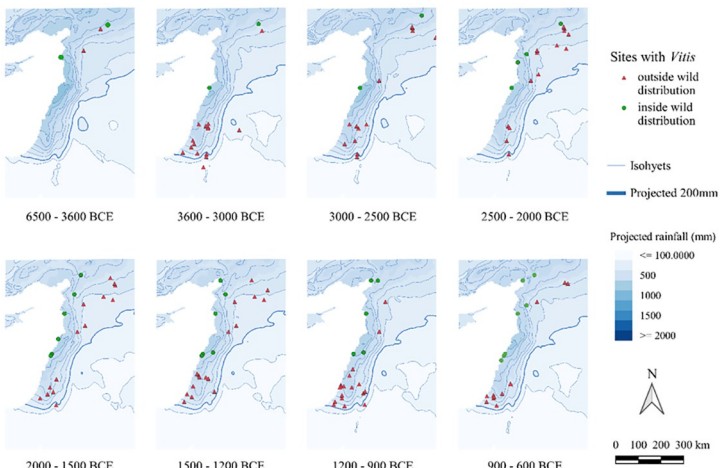

**Fig 7. Maps, with a particular focus on the Levant, of archaeological sites with *Vitis* finds, seeds or charcoal, indicating whether they fall within Zohary and Hopf's wild distribution [46].** The isohyets were calculated by the averaging of rainfall values hindcasted from Soreq Cave data within each respective period [50].

Fig 10 is a boxplot of the overall percentage results for the two botanical taxa in the northern and Southern Levant, regions where both grape and olive were intensively used. The boxplots for grape seed percentages look somewhat like those for olive, but the charcoal boxplots for olive and grape are very different, with overall very low values for grape charcoal.

To investigate the role of ancient rainfall in relation to the intensity of involvement with *Vitis*/*Olea* cultivation, the estimates of former rainfall were plotted over seed, seed ubiquity and charcoal percentages. Linear regression analysis was undertaken to investigate whether more rainfall correlates with more olive/grape cultivation/involvement, which can be expected if rainfall was a factor in deciding to invest in olive and/or grape. In this region, reduced rainfall acts as a strongly delimiting factor in plant growth (cf. [57] for likelihood distribution of olive according to rainfall). Reconstructed rainfall values were taken from [50] (cf. S1 Table). While at first sight the dots on Fig 11 may not disclose a pattern, there is a statistical significant correlation (p values <0.001) for the olive stone %, olive stone ubiquity % and charcoal % with rainfall, indicating higher olive percentages at sites with higher rainfall, although the linear

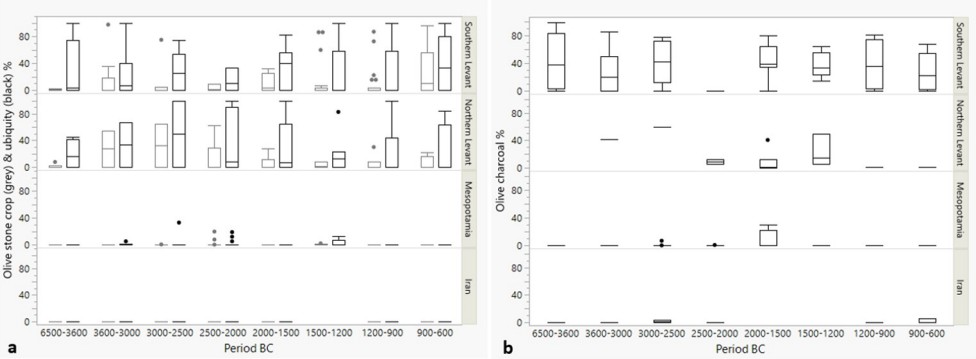

**Fig 8. Boxplot diagrams for *Olea* stone proportions (left/grey boxes) and ubiquity percentages (right/black boxes) (a) and charcoal (b) percentages per region and period.** See Table 1 for the number of sites/period and region.

**Table 2. Results of the Dunn's test on the olive and grape proportions.** Data as in S1 with "Region" and "Period BC" merged. Only group comparisons that delivered significant differences are provided with their mean score difference, standard error difference, Z- and p-value.

| | groups compared | | mean score difference | standard error difference | Z-value | p-value |
|---|---|---|---|---|---|---|
| olive stone % of crops | Southern Levant 900–600 BC | Mesopotamia 3000–2500 BC | 98.593 | 20.00393 | 4.92869 | 0.0002 |
| | Southern Levant 900–600 BC | Mesopotamia 6500–3600 BC | 103.135 | 23.79235 | 4.61405 | 0.0009 |
| | Southern Levant 900–600 BC | Mesopotamia 2000–1500 BC | 103.135 | 23.79235 | 4.33479 | 0.0034 |
| | Southern Levant 2000–1500 BC | Mesopotamia 3000–2500 BC | 81.571 | 21.73086 | 3.75367 | 0.0403 |
| olive stone ubiquity % | Southern Levant 900–600 BC | Mesopotamia 3000–2500 BC | 84.9792 | 20.04643 | 4.23912 | 0.0052 |
| | Southern Levant 900–600 BC | Mesopotamia 6500–3600 BC | 89.9667 | 22.23951 | 4.04535 | 0.0121 |
| | Southern Levant 900–600 BC | Mesopotamia 2000–1500 BC | 89.9583 | 23.58857 | 3.81364 | 0.0316 |
| grape seed % of crops | Southern Levant 6500–3600 BC | Northern Levant 1500–1200 BC | -149.038 | 32.55026 | -4.57871 | 0,0011 |
| | Northern Levant 1500–1200 BC | Mesopotamia 6500–3600 BC | 134.964 | 32.91807 | 4.10001 | 0.0095 |
| | Southern Levant 6500–3600 BC | Southern Levant 900–600 BC | -96.05 | 24.86067 | -3.86353 | 0.0258 |
| | Southern Levant 6500–3600 BC | Southern Levant 1200–900 BC | -87.063 | 23.05671 | -3.77603 | 0.0368 |
| | Southern Levant 6500–3600 BC | Northern Levant 2000–1500 BC | -108.017 | 29.03101 | -3.72073 | 0.0459 |
| grape seed ubiquity % | Southern Levant 6500–3600 BC | Northern Levant 2000–1500 BC | -112.417 | 26.53235 | -4.23697 | 0.0048 |
| | Northern Levant 2000–1500 BC | Mesopotamia 6500–3600 BC | 112.408 | 27.82736 | 4.03949 | 0.0112 |
| | Southern Levant 6500–3600 BC | Northern Levant 1500–1200 BC | -119.681 | 29.74870 | -4.02307 | 0.0121 |
| | Northern Levant 1500–1200 BC | Mesopotamia 6500–3600 BC | 119.673 | 30.90925 | 3.87174 | 0.0227 |
| | Southern Levant 6500–3600 BC | Southern Levant 900–600 BC | -89.667 | 23.73125 | -3.77842 | 0.0331 |

regression explains only ca 9% of the variation for the olive stone percentages ($F_{(1, 279)} = 27.5162$), compared to 20% of the olive stone ubiquity ($F_{(1, 279)} = 67.5188$) and 15% of the variation of the olive charcoal % over rainfall ($F_{(1,119)} = 20.2647$).

From Fig 11 it is visible that few sites with olive are found below 250 mm (reconstructed) annual rainfall. While that is the conventional dryland cultivation border, it is generally known that olive does not thrive and produce many fruits below 400 mm of rainfall [24]. Fig 12 depicts the sites below 400 mm (reconstructed) rainfall. Fig 11a appears to confirm this, since olive stone % of all crop plants higher than 40% occur only in settlements located in a region above the reconstructed 400 mm rainfall range.

Fig 13 depicts the grape seed and grape vine charcoal percentages over the reconstructed rainfall for the different sites. No significant linear correlation could be observed. Grape seeds and charcoals are represented at some sites with extremely low rainfall levels (below 250 mm).

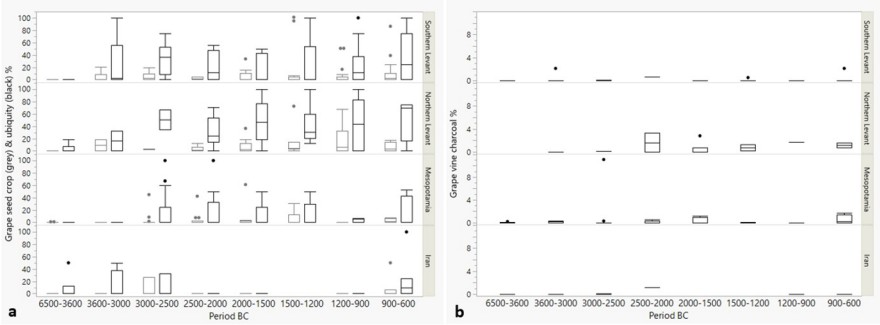

**Fig 9. Boxplot diagrams for *Vitis* seed proportions and ubiquity percentages (a) and *Vitis* charcoal (b) percentages per region and period.** See Table 1 for the number of sites/period and region. Note that the scale of 7a is exaggerated compared to Fig 7b.

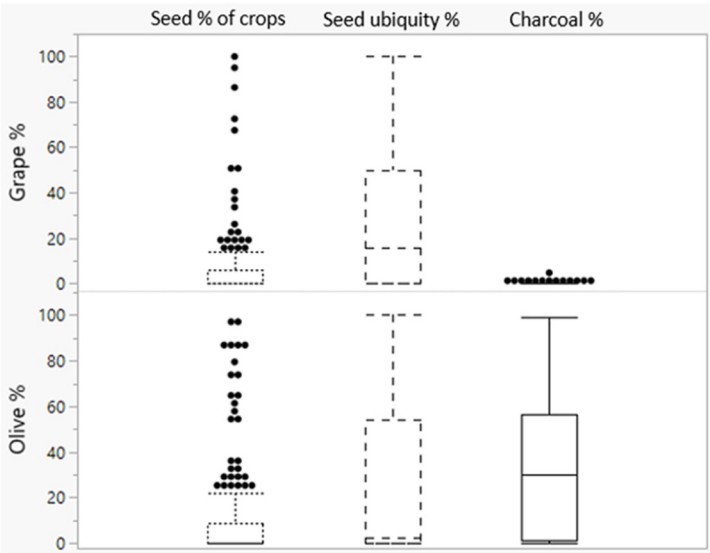

**Fig 10. Summary boxplots for the percentages of *Vitis* and *Olea* seed and charcoal remains in the Levant (N sites with grape seed % of crop = 175, N sites with olive seed % of crops = 162, N sites with grape seed ubiquity % = 163, N sites with olive ubiquity % = 157, N sites with olive charcoal % = 69, N sites with grape charcoal % = 65).**

The data show a trend of decreasing average rainfall between 5000 and 2500 BC for the sites with olive, with the lowest average levels between 3200 and 2300 BC. While the smoothed line for average reconstructed rainfall generally remains above or around approximately 400 mm for the whole period, in the late 4th millennium BC the smoothed line for minimal reconstructed rainfall values only reaches just above 250 mm. It should be noted though that the time spans for rainfall reconstruction in the earlier periods are larger than in later periods, increasing the likelihood of minimal rainfall in the earlier periods, possibly in part explaining the distance between average and minimal rainfall on Fig 14. After 2000 BC, sites with olive finds are again located in regions that received more rainfall on average (Fig 14).

The elevational distribution of sites with olive and grape over time show some similarities (Fig 15). While in earlier periods, especially between 6500 and 2000 BC for olive and 3600 and 1500 BC for grape, sites occurred quite equally in different elevations, with olive having the highest percentages at elevations 400–200 m, they afterwards dominantly appear in the zone between 0 and 200 m elevation.

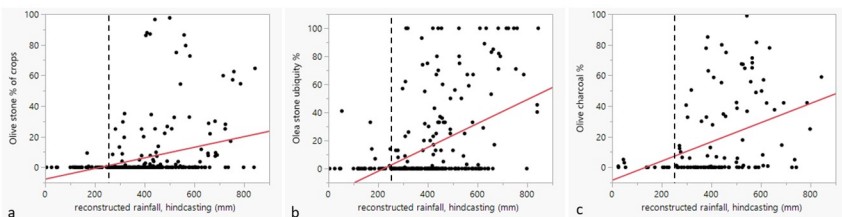

**Fig 11. Diagrams that show the relation between olive proportions (a), olive ubiquity (b) and olive charcoal (c) percentages and rainfall.** The linear regression line is indicated in red. Raised X-axis for visual understanding. Dotted vertical line indicates 250 mm rainfall.

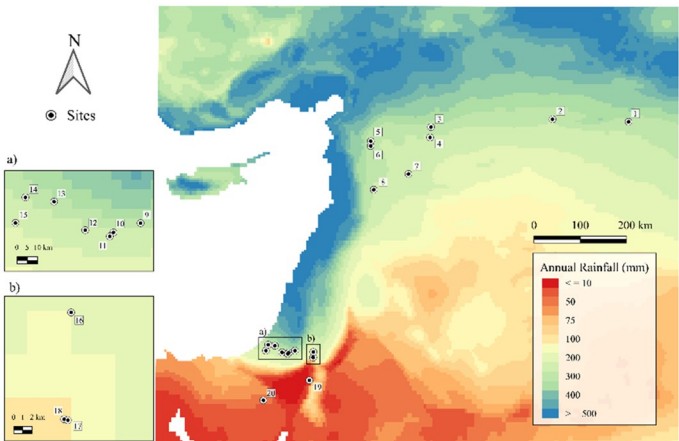

Fig 12. Sites with olive, seed or charcoal, that may have needed irrigation in case of local cultivation, i.e., sites with a hindcasted rainfall less than 400mm. All sites are labeled with numbers, the corresponding numbers follow: 1) Tell Taya, 2) Tell Bderi, 3) Tell Munbāqa, 4) Emar, 5) Tell Afis, 6) Tell Mardikh/Ebla, 7) Tell al-Rawda, 8) Tell Mishrifeh/Qatna, 9) Arad, 10) Tell 'Ira Khirbet el-Garra, 11) Hirbet el-Msas/Tel Masos, 12) Beer-Sheba/Tell es-Seba', 13) Tell Sera, 14) Qubur al-Walaydah, 15) Tel Farah South, 16) Bab'edh Dhra, 17) Numeira, 18) Ras an-Numayra, 19) Feinan, 20) Kuntillet Ajrud. Depicted is the modern rainfall as in Hewett et al. [50].

In Fig 16 the proportions for the different measures (% of crops, ubiquity % and charcoal %) for olive and grape are depicted over time, reconstructed rainfall, and elevation. The highest values for olive across the periods appear to fall within the range of 0–400 m of elevation (Fig 16a and 16b), particularly concerning phases post 1600 BC. As visible in Fig 15, the later phases (in green in Fig 16a) are proportionally strongly represented in regions between 0 to 200 m elevation that received (a reconstructed) 400 to 800 mm of rainfall. Ubiquity % of olive (Fig 16b) at the earlier settlements (in red tones) are high compared to later periods at higher and low elevations, such as the Jordan Rift Valley (red), the latter also reflected in the charcoal % (Fig 16c). The olive wood charcoal % seem however more diverse in terms of the distribution, elevation, and precipitation ranges.

Regarding grape, the largest values for grape seed % are located at an elevation between 0 and 500 m and the (reconstructed) annual rainfall between 400 and 800 mm (Fig 16d). Within that range several sites with high ubiquities % of grape seeds are found (Fig 16e). Notably, grape is represented at early sites in extreme positions, such as low rainfall (less than 250 mm) and high and low elevation (in red tones on Fig 16d, 16e and 16f). Grape charcoal appears poorly represented across the archaeological sites, but especially in zones between 0 and 500 m

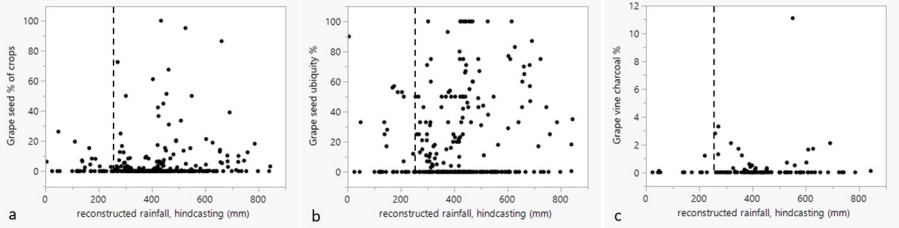

Fig 13. Grape seed (a) Grape ubiquity (b), and grape vine charcoal (c) percentages over (reconstructed) rainfall for the different sites. Raised X-axis for visual understanding. Dotted vertical line indicates 250 mm rainfall.

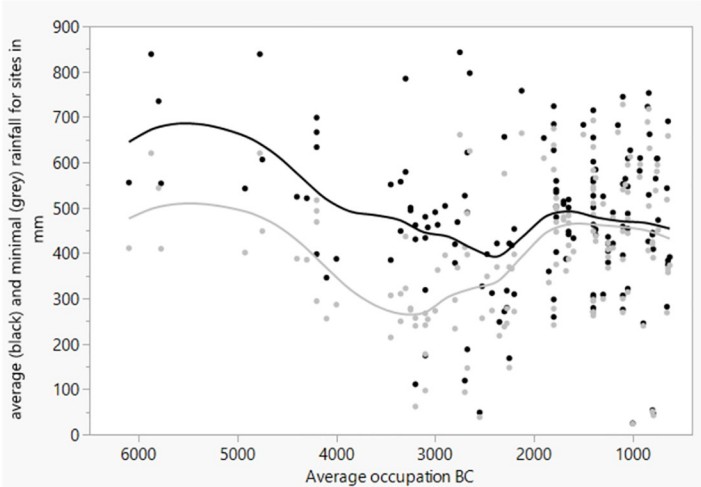

**Fig 14. Depiction of reconstructed average and minimal rainfall (respectively black and grey) over time for sites with olive finds (seeds and/or charcoal), with smoothed curves over the dataset using cubic spline with lambda set to 0.05 and standardized X values.**

some sites had a small proportion of grape charcoal, with a slight tendency for representation of earlier sites (in red) at somewhat higher elevations than younger sites (Fig 16f).

Fig 17 shows the results of the canonical correspondence analysis on major crops from 242 Southwest Asian sites with chronology as functional trait. Over 98% of the variation is explained by the axes. We can observe an increased focus on fruit tree cultivation through time. The Iron Age 2 (IA2 in Fig 17) from 900–600 BC stands out and could suggest a period of highly specialized fruit tree cultivation in the Southern Levant. The CCA also indicates other patterns, such as the cultivation of emmer wheat primarily during the Late Neolithic/Chalcolithic period (LN_C in Fig 17) (6500–3600 BC) and the Early Bronze Age (EBA1 in Fig 17) phases (3600–3000 BC) as well as the higher abundance of free-threshing wheat (Free-Wheat in Fig 17) from the Middle Bronze Age on.

The significance of olive and grape cultivation in the Levant merits further consideration. Fig 18 shows the Summed Probability Distributions (SPD) of radiocarbon dates (Palmisano et al. 2021) equating population density of the Levant alongside the olive and grape

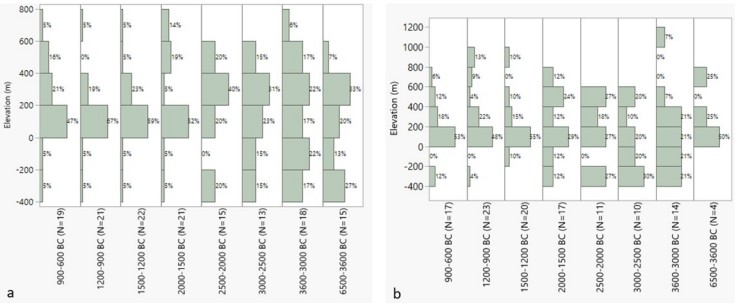

**Fig 15.** a. Elevational distribution of sites with olive presence (seed and/or charcoal) for the different periods of Southwest Asia, though mainly Levant), b. elevation distribution of sites with grape seed and/or charcoal for different periods in the Levant.

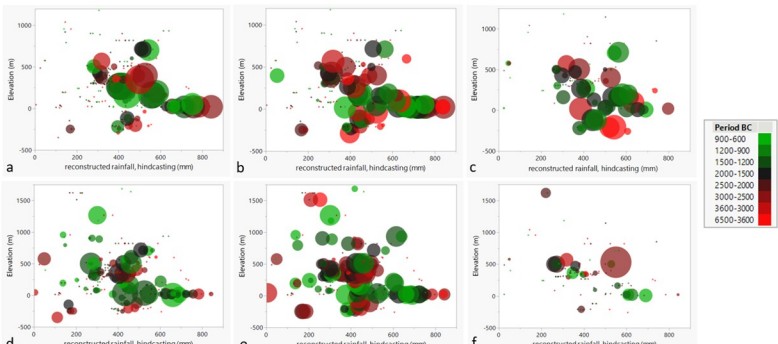

**Fig 16. a) Olive stone %, b) olive stone ubiquity % and c) olive charcoal %; d) grape seed %; e) grape seed ubiquity %; f) grape vine charcoal % reflected as circle size at sites, with indication of elevation (y-axis), over reconstructed rainfall (x-axis) and period of the site (colour).** Note that the dataset is somewhat different than for Fig 15 since all regions are included and no sites with lack of numeric data were depicted here.

archaeobotanical data. There are marked similarities between trends of olive and grape ubiquity, olive charcoal percentage and the occupation density graphs.

Over the long term, between 6000 and 900 BC, mean population density increased through time. However, population decline is indicated in the first half of the 4[th] millennium BC, after the 4.2 ka BP event, and between 1400 BC and 700 BC. The botanical data for the first half of the 4[th] millennium BC is limited, but the modeling of the preliminary data suggests a drop in olive stone ubiquity as well as in olive charcoal percentages (Fig 18). A drop in olive stone proportions, olive and grape ubiquity and olive charcoal percentages is also visible across the 4.2 ka BP event but started already around 2800 BC. Finally, a decrease in olive stone proportions and olive charcoal percentages occurs around the 3.2 ka BP event, just after a similar decline in the population curve (though note the temporal resolution of the archaeobotanical data is generally lower than that of the SPD). The 700 BC low in population is not visible in the archaeobotanical data, showing increases in grape and olive seed proportions and ubiquities and olive charcoal percentages.

Population increase appears to have taken place in the second half of the 4[th] millennium BC and during the 3.2 ka BP event. Starting to increase in the second part of the 4[th] millennium BC, highest ubiquity percentages are reached for olive stones, olive charcoal and grape seeds

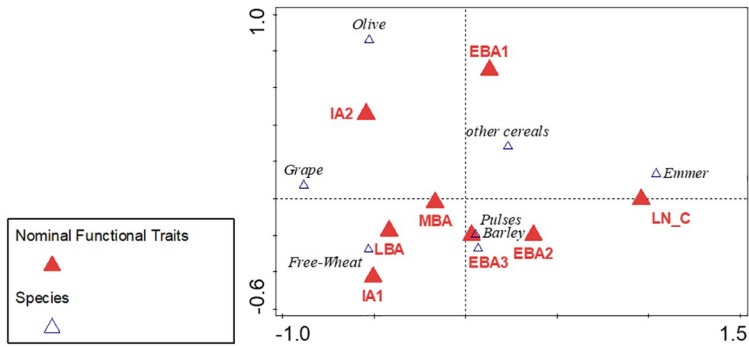

**Fig 17. Canonical correspondence analysis on major crops (n = 242 sites) with chronology as functional trait, with LN_C (6500–3600 BC), EBA1 (3600–3000 BC), EBA2 (3000–2500 BC), EBA3 (2500–2100 BC), MBA (2100–1650 BC), LBA (1650-1200BC), IA1 (1200-900BC), IA2 (900-600BC).**

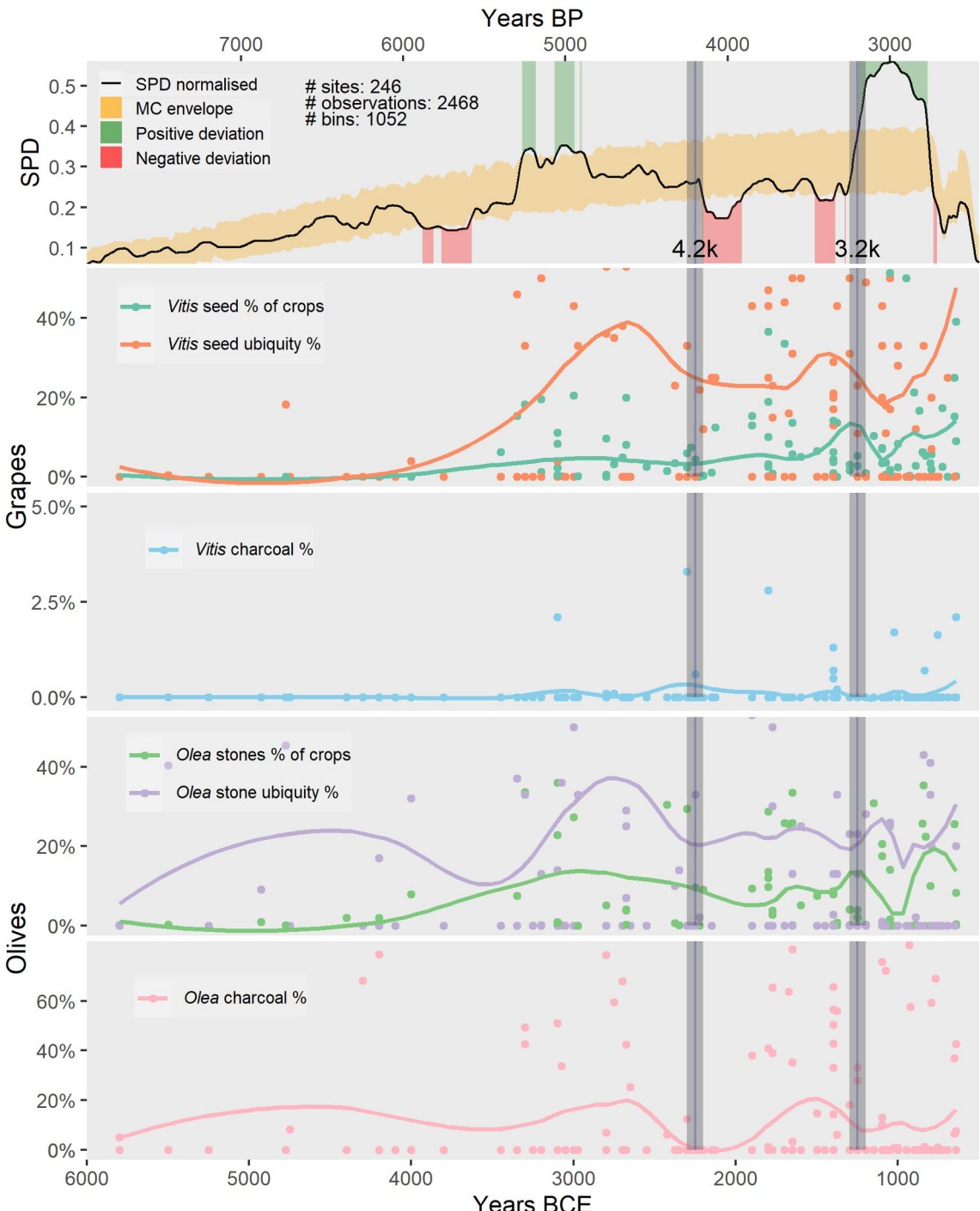

**Fig 18. Summed probability density (SPD) distribution of 2468 radiocarbon dates from 246 sites in the Levant, with a 95% confidence envelope created through Monte-Carlo methods (here the MC envelope).** The olive and grape % lines depict smoothed curves using loess (Locally estimated/weighted Scatterplot Smoothing), with a span of 0.3 and a confidence level of 0.95. for grape/olive seed %, grape/olive ubiquity %, grape vine/olive charcoal % for all available sites from the Levant (see data S1 Table). Y axes scales have been optimized to show curves. The archaeobotanical results are depicted on the chronological axis (X-axis) as a point in the middle of the occupation span.

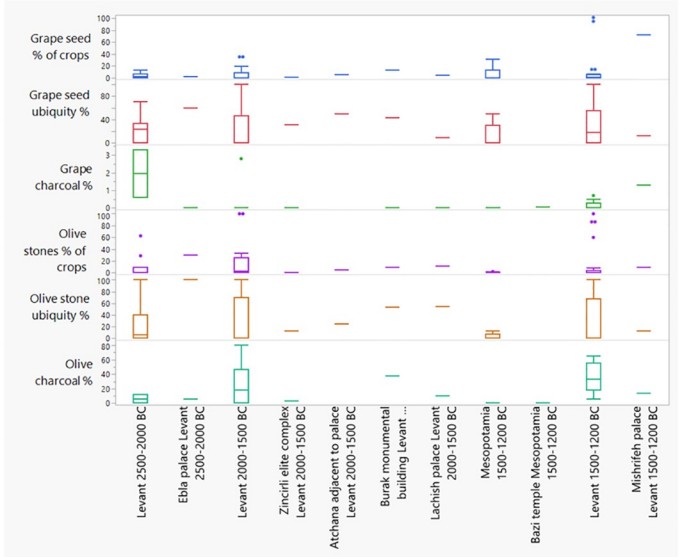

**Fig 19. Comparison of palace *Olea/Vitis* seed/charcoal percentages with those from mostly domestic contexts in this period and region.** Note that for Lachish different values were used compared to S1 Table, because here only palace seeds and charcoals, are depicted [58, 59]. Note that the X-axes for *Vitis* charcoal % was raised and that Y-axes scales differ.

shortly before the mid of the third millennium BC. After the 3.2 ka BP event, olive stone ubiquity percentages increase again, though at a much smaller scale than population increase, but grape first decreases and seems to only increase after 800 BC.

To investigate elite consumption, storage, or institutional control of olive and grape production, the *Olea/Vitis* seed and charcoal percentages from elite buildings, such as palaces, temples, and monumental buildings, are compared with those from mostly non-elite buildings from that region and period (Fig 19). The finds from most elite buildings fall within the range of contemporary sites in the region, though above the median for seed ubiquity %. The Tell Mishrifeh/Qatna palace (1600–1200 BC) stands out for its high *Vitis* seed and charcoal percentages. Also, its olive seed percentages (of crops) are on the high end compared to other sites in the region of this period, but not its olive stone ubiquity percentages and olive charcoal percentages. The Ebla (Tell Mardikh) palace G (from the phase 2400–2300 BC) has a high percentage of olive stones compared to other sites in the Levant of that period, though has an about median value for olive charcoal percentages. Also, its *Vitis* seed ubiquity percentages are quite high. The Burak monumental building, dating to the phase 2000–1800 BC, also had rather high *Vitis* seed percentages amongst the crops.

## Discussion

### Earliest indications for olive and grape cultivation in Southwest Asia

Earliest cultivation typically begins in areas within the natural distribution range of the wild progenitor of crop species [60]. Assessing earliest cultivation in the archaeobotanical context is problematic because seeds can hardly be differentiated from having been cultivated or simply collected from the wild. Although phytogeographic changes happened throughout the Late Pleistocene and Holocene, archaeobotanical findings of potentially cultivated fruit seeds

outside their natural distribution ranges can be an indication of cultivation, especially when their wood charcoal is also present assuming wood was likely locally collected.

## Start of olive cultivation

The earliest indication for olive cultivation is claimed by the Southern Levant and dates to ca. 5000 BC from Tel Tsaf [45] and Tell esh-Shuna, where olive wood charcoal and seeds have been found outside the natural distribution mapped by Zohary and Hopf [46]. Notably, this evidence predates urbanization in this region. For the Northern Levant, the earliest evidence of olive cultivation outside its natural distribution is much later, from the phase 2500–2000 BC. Apart from the general limitation of fewer archaeobotanical data sets in the Northern Levant, there is also a lack of archaeobotanical data for sites outside the habitat of oleaster for the earlier phases, perhaps explaining the discrepancy between the archaeobotanical and genetic evidence, where the latter indicates a domestication center of olive in the Northern Levant [8]. In addition, compared to the Southern Levant the Northern Levant is characterized by more areas of higher elevation unsuitable for olive growth (cf. SRTM 1 arc-second data, U. S. Geological Survey), which further minimizes the chance of finding olive in sites away from its natural distribution range, thereby making it less likely to detect early cultivation.

## Start of grape cultivation

Detecting possible early cultivation of grape outside its wild habitat using Zohary and Hopf's map [46] proves to be even more challenging. While this map suggests Tell Brak, where grape seeds and charcoal were found dating to 4000–3600 BC, lies outside the natural distribution range of wild grape vine, recent research indicates that this may not have been the case [16]. There are also other early finds of *Vitis* seeds from that region (cf. Figs 3 and 6), for example at Tilbeshar (5400–3100 BC) and Grai Resh (5th/4th millennium BC) [61, 62]. The find of under-developed grape seeds at Kurban Höyük dating to the end of the 4th millennium BC also suggests grape cultivation [63] and fits with the results of morphometric analysis which identified domesticated grape by the latest in the 3rd millennium BC in northern Mesopotamia [16].

*Vitis* seeds are also found as far east as Tepe Yahyah in southern Iran from the phase 4500–3800 BC (Figs 3 and 6). The early suggestion of wine residues in pottery from northern Iran, within the natural distribution, dating to 5400–5000 BC, may perhaps indicate cultivation [64], matching the evidence from Naqinezhad et al. [47] showing a larger natural distribution than Zohary and Hopf [46].

The first evidence of both *Vitis* seed and charcoal presence in the Southern Levant seemingly outside present-day distributions are from Arad dating to the period 3600–3000 BC. Arad's reconstructed annual rainfall in this period is only 320 mm (S1 Table), which is quite low. There are several other sites from this period in the Southern Levant with grape pip evidence, but so far, no investigated charcoals, in zones that would have received extremely low rainfall levels (Fig 3). For example, Ras an Numeira (3350–2910 BC) has a reconstructed annual rainfall of 110 mm and has evidence for carbonized grape pressings, perhaps indicating local grape cultivation using irrigation [65]. There are also other water-demanding crops, such as flax, reported from the site. The grape remains there seem to have had a combination of wild and domestic traits [65]. Hence, there exist several indications that grape was cultivated in the Southern Levant by around 3600–3000 BC.

In the Northern Levant at Ras Shamra, though within the region of wild grape, grape seed finds are present from phase 6500–5250 BC at a very low ubiquity, increasing in ubiquity from phase 5250–4300 BC onwards (to ca. 18%), perhaps indicative of an increased involvement

with grape and cultivation and though no charcoals were investigated, the seeds equally may represent remains of wild grape utilization.

In all, strong archaeobotanical evidence for *Vitis* cultivation in Southwest Asia is much later than the 8000–6000 BC Caucasian evidence [66–68] and the genetic inference date for Levantine domestication ca. 9000 BC [14], since grape starts to become well represented at archaeological sites only around 4000 BC in the Levant, with the possible exception of Ras Shamra. At this stage, the archaeobotanical data seems to favour the Caucasian first domestication model and later expansion of grape cultivation/domestication to Southwest Asia. However, there is a massive lack of data for the Northern Levant that may be crucial for our understanding of the processes. Recent morphological evidence from grape pips suggests that there was a hybrid of Asian and Caucasian domestic grape in northern Mesopotamia already by the 3rd millennium BC [16].

## The role of climate in expansion and contraction of olive/grape cultivation

The role of climate in the expansion and contraction of wild olive/grape vine has already been mentioned regarding their potential former wild distribution [69]. *Vitis* occurred further inland in the Levant during the Epipalaeolithic [70] and *Olea* distribution was likely very dynamic over the Pleistocene [8]. While climatic change in the last 8500 years was less extreme than at the Pleistocene-Holocene transition and before, it still likely had an impact on fruit tree cultivation, especially since cultivation was pushed beyond its natural habitat into areas where small fluctuations in rainfall or temperature would have had major impacts on yields and tree survival (cf. Fig 14 especially the period between 4000 and 2000 BC).

## Rainfall

Rainfall appears to have had some impact on the scale of involvement with olive cultivation for the period under consideration. Correlation tests indicate a low to modest correlation of olive with rainfall. This tendency to depend on rainfall for olive cultivation may explain the lack of olive cultivation in Iran for the considered period, where only through irrigation good yields can be achieved in the southern Zagros foothills, cf. Fig 4 in [71]. However, there is generally not much data from this region regardless.

Less clearly related to rainfall are decisions on the degree of involvement with grape cultivation, which is logical considering grape is naturally closer associated to riparian environments. Except for Konar Sandal, there is currently no evidence for *Vitis* wood below the 200 mm rainfall (reconstructed), and although this may be due to a small sample size and handpicked samples, it should be noted that there were several sites in those arid areas with *Vitis* seeds, including very early sites. Some of these were likely locally grown, since possible pressing remains were found, e.g., at Numeira (3350–2910 BC), which the authors suggest was irrigated [65]. In addition to grape, other crops with high water requirement, such as flax, were found there [65]. The overall lack of correlation of *Vitis* seeds and charcoal with rainfall may indicate that *Vitis* could more easily be accommodated in local riparian environments such as river valleys, or that irrigation was more commonly used for *Vitis* cultivation than for *Olea*. Of the 18 sites with grape below 250 mm reconstructed rainfall, 9 could be confirmed being located near important major water bodies (such as the Khabur, Euphrates, Dead Sea or Besor stream), whereas 6 are located along a dry wadi (e.g. Wadi Fidan, Konar Sandal, Numeira, Jawa and Zahrat adh-Dhra 1). The rainfall level for Khirbet al-Mudayna (el-'Aliya) was likely underestimated due to the small-scale local microclimate that was not captured in the present-day rainfall map.

The archaeobotanical evidence for the Levant shows marked percentage decreases for grape and olive, especially in seed ubiquity and charcoal, starting around 2800 BC and reaching a low around the 4.2 ka BP aridity event (Fig 18). Around the 3.2 ka BP aridity event grape seed ubiquity %, olive stone % of crops and olive charcoal % again decrease, as well as olive stone ubiquity % shortly after (Fig 18). Both lows during these times suggest a possible aridity impact on olive and grape cultivation. The typical dependance of olive cultivation on rainfall, and its extension somewhat away from its natural distribution especially between 4000 and 2000 BC (Fig 14), makes it prone to increased aridity which would particularly decrease in yield [24]. Since grapes are sensitive to drought stress [72], water scarcity may have affected grape production as well, despite certain precautionary measures, such as irrigation in the generally more arid regions. The 4.2 ka BP event not only likely impacted on grape, olive and other crop species [73, 74], but also seems to have had a negative effect on population density in the Levant (Fig 18). The less precise temporal resolution of the archaeobotanical data compared to the population density data makes it impossible to unravel the complex causal chains; despite this, aridity seems to have played a factor, with a decline in olive and grape and harvests possibly impacting caloric provisioning (compare with [75]) and disrupting trade and exchange networks in which olives, grapes and their derived products were likely very important [76, 77].

Interestingly, unlike the 4.2 ka BP event, the 3.2 ka BP was not associated with decreasing population density in the Levant, indicating that the dynamics were more complex and was probably compensated by the increasingly sophisticated networks that might have allowed compensation during locally occurring resource bottlenecks. The noticeable population increase around that time may relate to migrations from other regions for which there is ample evidence for the Levant [22]. In the Ugarit hinterlands textual evidence documents the destruction of vineyards through enemy attacks [22]. Employing destructive forces with the goal to weaken opponents might have contributed to strong differences in the agricultural resource situation between regions. Around this time, palace institutions that previously stimulated wine and olive production decline and there appears to have been a reorientation of production, sometimes described as ruralization [38]. Adaptive strategies were likely developed; for instance, pastoral products in some cases had greater contribution to the overall food supply. At Kinet Höyük in the Northern Levant, for example, destruction occurred around 1150/1130 BC after which the site was occupied seasonally, likely by pastoralists [78]. Simultaneously, there was a steep decrease of involvement with olive compared to the previous period (as seen from the charcoals; S1 Table). Such diversification in practices is also reflected in Figs 8 and 9. About half of the Late Bronze Age Levantine sites have no stones, as taken from a median around 0, while the other half were still involved with olive, whereas in the previous period proportionally more sites were involved with olive (e.g., for the phase 2000–1500 BC in the Southern levant or 1500–1200 BC in the Northern Levant).

## Temperature

Olive and grape are both cold-sensitive and cannot withstand strong frost. This is also why the northern Zagros mountains are unsuitable for olive cultivation [71]. Grape vines can be damaged by temperatures around -15 and -20 ˚C [54], while olive is not able to survive below -12˚C and severe frost damage occurs already at -7˚C [79]. Kaniewski et al. [24] have shown that the optimal mean annual temperature for olive is 16,8˚C and that temperature changes have affected olive growth and yields in the Tyre region of the Levant through time.

The progressive cooling between 2500 and 0 BC could have especially impacted the occurrence of *Vitis* and *Olea* at higher elevations, potentially leading to a reduction in their

occurrence in those areas. Accessibility of land in these areas may also have played a role in this. The trend towards a concentrated presence of olive and grape between 0 and 200 m begins for olive from 2000–600 BC (Fig 15a) and for grape from 1500–600 BC (Fig 15b).

## Arboricultural practices

**Irrigation.** Olive production was mainly a focus of the Northern and Southern Levant in the period under consideration (Figs 4, 5 and 8). There it was exploited mostly (but not exclusively) in zones where dry farming was possible. The weak correlation between the archaeobotanical record of olive and rainfall suggests that irrigation also played a minor role in olive management practices. Moreover, some sites across our regional categories and periods fell below the 400mm limit for olive yield, of which several had additional (sometimes small-scale) evidence for irrigation, such as Rawda (2500–2000 BC) [80, 81], Tell Mishrifeh/Qatna (2500–2000 BC, 1500–1200 BC) [82], Emar (2500–2000 BC) [83], Arad (3600–3000 BC) [84], Deir'-Alla (phase 900–600 BC) [85, 86], Lachish (phase 2000–1500 BC and 1500–1200 BC) [58].

Overall, the evidence seems to indicate that although much of the olive cultivation took place within its optimal range of rainfall, at sites where irrigation and water management practices and infrastructure were in place, olives could be accommodated and particularly in conjunction with other water intensive crops such as flax and grape. Grape cultivation appears to have been less dependent on natural rainfall compared to olive, despite the fact that grape cultivation requires 500–1,200 mm soil moisture during the grape growing season (February to July in Southwest Asia) to produce good harvests (FAO Land and Water Development Division, http://www.fao.org/landandwater/default.stm). This may therefore indicate that irrigation was more regularly applied to grape, either directly or indirectly by taking advantage of grape plantations above easily available aquifers or by managing water resources in rocky wadi valleys. Multiple passages in the Hebrew Bible mention planting grape along the river (e.g., Ez 17,8; 19,10) and some Hittite texts also refer to irrigation of grape vine [87].

## Differences in pruning practices *Olea* vs. *Vitis*?

Another difference in the management of grape and olive may relate to the degree of pruning that took place. For olive, pruning is beneficial for the support of the fruit load, to gain more exposure to sunlight, to rejuvenate the tree and to reduce pests [88] while for grape it is especially useful to maximize the quantity of berry clusters [63]. In our dataset, *Vitis* charcoal percentages are generally extremely low compared to *Olea* (ubiquity) percentages (Fig 10), even though the mass of pruning waste of grape and olive per hectare is similar [44]. The low charcoal percentages of *Vitis* compared to *Olea* are present even at sites with evidence for large scale exploitation of grapes.

A possible explanation could be that historic pruning of grape vine was less intensive than that of olive. This may relate to the growing of "wedded" vines (i.e., vines cultivated attached to a tree), evidence for which is found in the Hebrew Bible (e.g., 1Kön 5,5; Mi 4,4; Sach 3,10), Hittite texts and Neo-Assyrian iconographic sources [87, 89]. This practice is also ethnographically known from the Levant [90]. Wedded vines are more difficult to prune, which may have reduced pruning when compared to modern intensive grape cultivation. Some ancient sources indicate that pruning was applied, such as Neo-Assyrian period iconography [89] and Hittite [87] and Old Testament (Iron Age) texts [91], the latter even mentioning pruning remains being used as fuel [91]. Another possibility is that vine pruning waste was used for other purposes than fuel, such as for basket weaving, or as fertilizer after composting [92]. A Hittite text even refers to smaller vine wood served for lighting the inside of buildings, as opposed to wood from trees used as heating [87]. Such probable variations in the use of vine wood may

have implications on preservation, but also the charring process for grape vine twigs compared to olive twigs may have different effects, because the average diameter of olive twigs is likely to have been somewhat larger than of grape vine. Small diameters could have been fully combusted into ashes. Personal experience has shown that at some sites *Vitis* charcoal tends to be vitrified, making it difficult to identify, also causing some underrepresentation in the archaeobotanical record.

## Socioeconomic roles of olives and grapes

While the history of olive and grape cultivation was partially shaped by climate and environment, there are also striking resemblances between the population density curve and the olive/grape curves of the Levant, suggesting the importance of these crops for sustaining communities, determining food security and for contributing to the economy through surplus production, trade, and exchange (see also Fig 18).

## Increased social stratification in the 4th millennium BC Levant

During the second half of the 4th millennium BC a notable surge in population occurred in the Levant, during which there is also an increase in ubiquities of olive and grape at archaeological sites. This suggests that, amongst possibly other reasons, olive and grape cultivation impacted the population growth in the region or were at least somehow related to increasing population density. The trends show a relatively abrupt increase in population numbers, whereas grape and olive feature a slow increase, reaching a peak only after population again decreases. This may point to human action to increased demand for olive and grape, perhaps as a reaction to raising population numbers. The cultivation of these crops provided viable and calory-rich food products, that could be grown in areas that were unsuitable for agriculture, such as hilly and rocky terrain, expanding the total amount of exploitable land for crops and therefore the capacity to feed a larger population. This expansion of olive cultivation has been documented in detail for the lake Galilee catchment around the late 6th millennium BC, involving fire to clear new arboricultural land [93]. The expansion is also visible in the distribution of sites with olive into higher and lower elevations (Figs 2 and 16a–16c). The exploitation of marginal lands for arboriculture seems to have been even more pronounced for grape, since some of the early sites with grape in the Levant were in areas that received very little rainfall (Fig 16d–16f), needing irrigation to allow cultivation. Terracing may also have played a role here [94], enabling both extensification and intensification of production.

Olive and grape were not only important staple products but also offered economic opportunities through surplus production and trade as cash crops, especially since olive and grape can be transformed into products that do not decay easily, are easy to transport and for which there was local and regional demand [3]. Ceramic studies have suggested that wine and olive oil were among the materials that were imported into Egypt from the Levant, likely already starting from the end of the fourth millennium and then developing throughout the third millennium with the entire Levant being involved in the exchanges. Principle economic partners changed through time, with most of the materials likely being imported first from the Southern Levant, while later the Central and Northern Levant assumed a privileged role [95–100]. These developments likely contributed to supporting the increasing social stratification, which becomes already evident in the urbanization processes during the first half of the third millennium. In fact, from around 3000 BC onward, together with the involvement of the Levantine elites in the interregional exchange with Egypt, there is increasing evidence for palace institutions and industrial production in the Levant, indicating the centralization of resources [17, 31, 101, 102].

## Olive and grape in palaces and elite buildings

Most Levantine palaces and monumental buildings show higher than median percentages for grape and olive seed ubiquity compared to respective periods and regions (Fig 19). Furthermore, regular finds of olive charcoal in Levantine palaces and monumental building contexts suggest institutional engagement in olive cultivation, especially in cases where it was present in waste deposits which was for example the case in the palace of Mishrifeh/Qatna room DK which contained large volumes of kitchen waste [59]. Strongly represented alongside olive charcoal there was also grape vine charcoal [59]. While contextual analysis of the charcoal samples from the Lachish palace remains is ongoing, here also olive charcoal was present in waste dump material associated with the palace. Additionally, of note is that Qatna, Lachish and Ebla all would have required irrigation for local olive cultivation. While the analysis of Bronze Age texts particularly stress the role of the palace in controlling and managing olive and grape revenues through taxation of the rural population [4, 5] and the allocation of olive groves and vineyards to elite members [6, 103], Monroe [104] recently highlighted that there is also textual evidence for grape production on royally owned land for the 13th/12th century BC (e.g. at Emar in Mesopotamia and Ugarit in the Levant), besides the much larger proportion mentioned to be in private hands, mostly paying taxes. The charcoal data for the palaces at Mishrifeh/Qatna (1600–1200 BC) and Lachish (1800–1550 BC) further support such institutional engagement in cultivation. This could imply less of a market-oriented system and an economy not solely reliant on tax funding.

While there is very limited evidence for grape charcoal from the Tell Bazi temple in the Middle Euphrates (1800–1350 BC), perhaps suggesting some engagement in grape cultivation, there is no evidence for olive cultivation, which may in part reflect local preferences and adaptations. Olive appears not to have played a major role in the Mesopotamian diet and its cultivation was probably also limited to some ecologically well-suited regions (Figs 4 and 8). It was overall less prevalent in the botanical remains, and texts indicate olive oil was only imported in modest quantities largely for cosmetic products [105].

## Trade and import

In contrast to Mesopotamia, Egypt was a major importer of olive oil during much of the period under consideration (4.2), with this trade playing an important role for the economic and political integration of the Southern Levant (4.1). While the routes and format of the contact changed over time, there is evidence for intensive exchange with Egypt after the 4th millennium BC, extending to outright political control [106]. Olive oil and wine trade in the ancient Levant are frequently discussed as a combined endeavor [106], though proving in any specific case is difficult and even techniques utilizing chemical biomarkers or residue analysis struggle to distinguish between different vegetal remains [107]. However, the combination of textual evidence and pottery can help us gain insight into long-distance trade networks, such as movement of wine from the Northern Levant to Mesopotamia in the 19th-17th centuries BC [7]. From the second millennium BC onwards, the scale of trade from the Levant to areas such as the Aegean increased [106]. The cargo of the Uluburun shipwreck, which sank off the coast of Turkey around 1300 BC, contained international trade ware including "Canaanite jars" (storage jars from the Levant) and the remains of olive and grape (wine and or raisins) [106, 108]. By that time, wine was a commodity and no longer a luxury as documented in textual sources, with one liter in the 1300s costing less than a daytime's work compared to approximately 2 months' work in the early 2nd millennium BC [104]. This was likely in part related to reduced transport costs, especially associated with improvements in sailing and maritime transport [104] and the emergence of the merchant class [109].

Trade of olive oil and wine are far more intensively discussed in publications than raisins, olives, and olive wood. Olive consumption is known from at least the mid-5th millennium BC [110] and the Uluburun shipwreck contained a storage jar full of dried olives which may have been a trade product or for on-board consumption [108]. Our archaeobotanical review indicates only two sites with olive stones that did not have wood, which we might interpret as sites importing olive fruits, Kuntillet Ajrud (900–600 BC) and Emar (1500–1200 BC), both of which are located outside the dry farming limits for olive. Overall, the number of sites that likely only imported olive as a fruit are very small, therefore we can confidently say that olive-related trade focused on olive oil (Fig 2).

Conversely, there are 7 sites with evidence for olive wood but no olive seeds (Fig 2). In the case of Konar Sandal, the wood was likely from another wild olive subspecies. The lack of olive stones may be due to a restricted number of seed samples or to off-site olive processing. Alternatively, it may indicate the import of olive wood or, more likely, of artifacts made of olive wood. Tell Mozan in Northeastern Syria was intensively investigated archaeobotanically but the contexts of the time-block 2000–1500 BC contained no olive seeds, while olive charcoal made up 30% of the assemblage across 3 different contexts [111]. This is striking and merits further in-depth examination.

In our dataset, there are 38 sites with grape seeds but no grape vine wood, perhaps indicating a more distant location for grape yards or importation of raisins, especially likely in very arid regions (Figs 3, 16a and 16b). The underrepresentation of grape vine wood, possibly due to non-intensive pruning, likely also plays a role. For example, grape seeds but no wood charcoal were recovered in the Tell Mardikh/Ebla Palace in the 2500–2000 BC phase despite contemporary texts providing evidence for local cultivation along with the importation of wine [101].

There were also 7 sites that had only grape wood charcoal remains, though in very small proportions. Grape vine wood is generally considered of low value for woodworking but was traditionally used for basket weaving.

## The role of the neo-assyrians

From the late 8[th] to early 7[th] centuries BC imports from the Aegean and Western Mediterranean in the Levant decreased while Assyrian artefacts began to steadily rise [106]. This relates to the progressive expansion of the Neo-Assyrian empire that by the 9[th] century BC included the Northern Levant and from the 8[th] century BC parts of the Southern Levant [19, 38]. While it has been argued that economic interests, including access to olive oil, may have played a role in this expansion, cf. [19] versus [20], there are regional differences visible in the archaeobotanical remains as was previously indicated within the Southern Levant [21]. For instance, the Phoenician region (e.g. Burak and Sidon, here classified within the Southern Levant) shows the strongest evidence for intensive olive involvement but there appear low levels of involvement with olive in the Northern Levant, which entirely focuses within and near wild species distribution. The low olive values in the northern Levant would argue against a general Levantine Neo-Assyrian "olive-oil-exploitation-expansion hypothesis", which accords with the longue durée pattern that olive oil was not favored by Mesopotamians for food (4.2 and 4.3). Specialized production of olive in the Phoenician region was likely oriented towards local use and maritime trade westwards [20].

While the data suggests that olive was not the major direct interest of the Neo-Assyrians, there is evidence for grape in northern Mesopotamia and beyond, including the Neo-Assyrian empire in Iran. Depictions of grape vine in Ashurbanipal's palace convey the importance of grape to Neo-Assyrian kings [74, 89], as well as mentions to vines as gifts between the king and

family members [112]. With the promise of "a land of grain and wine, a land of bread and vineyards, a land of olive trees and honey" [113], the large-scale Neo-Assyrian irrigation projects and displacement of people to the heartland may have also played a role in the observed expansion of grape cultivation [114], although our results suggest that deportees anticipating olive may have been disappointed.

The canonical correspondence analysis for major crops (n = 242 sites) with chronology as functional trait shows a trend of increased specialization in olive and grape cultivation over time (Fig 17), with the period from 900–600 BC standing out. This is related to southern Levantine specialization in olive and grape cultivation, also regionally diverse within that region, cf. also [21]. This regional specialization was likely also a consequence of the expanded trade networks associated with the Neo-Assyrian territorial empire (compare with e.g., [115]) and increased specialization in trade of particular groups [21], hence a result of greater cultural integration. In the Northern Levant, particularly at Zincirli, a distinct pattern of specialization in animal exploitation emerged after incorporation into the Neo-Assyrian empire [116].

## Conclusion

Through our review study we shed new light onto possible areas that featured early olive and grape cultivation. We also gained new insights into the role of climate through time in the expansion and retraction of olive and grape cultivation and the implications this would have on human populations. Additionally, we acquired understandings into arboricultural practices, cultural preferences, trade, specialization, and the role of the palace in fruit tree cultivation. Here we summarize our findings into a diachronic narrative of the history of olive and grape cultivation.

Based on the current evidence, olive cultivation appears earlier than grape cultivation in Southwest Asia. Since there is indication for olive cultivation outside the distribution area of wild olive around 5000 BC in the Levant, the earliest cultivation likely occurred before that date and likely in the Northern Levant where our data is currently very limited. Olive cultivation and its expansion predate urbanization in this region by ca. two millennia. From the seed remains at archaeological sites, it appears that the scale of olive production in the Levant increased in the second half of the 4th millennium BC.

The earliest indications for grape cultivation appear in the 4th millennium BC in Southwest Asia. While earlier finds seem to focus around the northern Fertile Crescent and Iran, in the Southern Levant grape suddenly starts to appear in the phase 3600–3000 BC and occurs even in very arid regions where cultivation would have required intensive water management.

In the 4th millennium BC, increasing olive and grape production is evidence for favorable ecological and/or socio-economic conditions that boosted and/or followed population growth. An increased surplus food production may have also promoted social stratification, such as elite formation and centralization through e.g., trade, especially with Egypt.

The weak correlation between the degree of involvement with olive and rainfall in Southwest Asia suggests most olive production relied on dry farming. However, irrigation was likely practiced at some major centers, such as for the 3rd millennium BC Tell Mardikh/Ebla. Intensification there may have been both sponsored and controlled by the palace institution. Olive cultivation was increasingly pushed to more arid and higher elevation locations in the 4th and 3rd millennium BC, making the socio-economic system vulnerable to aridity and colder conditions.

The degree of involvement with grape cultivation in Southwest Asia appears to have not been correlated with rainfall, suggesting a more regular application of irrigation or cultivation

in river valleys. Besides likely having been more regularly irrigated compared to olive, grape received less pruning, perhaps sometimes because of the cultivation of "wedded grape".

A decline in olive and grape production and consumption starting shortly after 2800 BC, reaching minimum values ca. 2200 BC, was followed by a population density decrease beginning around 2600/2500 BC which, while first stagnating, then abruptly dropped shortly after 2200BC. While the causality is difficult to unravel since the archaeobotanical datasets maintain lower chronological precision than the population density dataset, the correlation suggests a complex interrelationship of diverse processes in which climate played a role through increased aridity. Such aridity, with a peak around 2200 BC and a longer phase of cooling, likely first affected crop yields and with food limitations followed population decrease, which then decreased labour input for fruit tree cultivation, in turn impacting trade and elites. From 2000 BC/1500 BC onwards, high production sites of olive and grape accumulate closer to the coastal zone at elevations between 0–200 m and are associated with the intensified maritime trade of olive oil and wine, which is archaeologically documented for much of the second millennium BC. Levantine palace institutions, such as at Burak and Mishrifeh/Qatna, further appear to have been involved with fruit tree cultivation.

By 1200 BC another period of settlement disruptions occurs, but this time, the Levantine population appears to have increased, possibly because of immigration driven by aridity, which is sometimes argued as a contributing factor to the observed changes. There is evidence for destruction of vineyards in the Northern Levant during this crisis. After the turmoil of 1200–900 BC, olive and grape did not regain its important status in the Northern Levant as different modes of subsistence, such as pastoralism, were adopted in regions where there was previously greater emphasis on olive and grape cultivation. In the phase 1200–900 BC in the Southern Levant, the importance of grape begins to increase, with olive regaining some importance.

By the 9th and 8th centuries BC respectively, much of the Levant was incorporated into the Neo-Assyrian empire. A strong decrease in population density can then be observed in the Levant, probably related to population deportations to other regions. While olive oil exploitation does not seem to have been the major direct interest of Neo-Assyrian rulers, in the Phoenician region including the Southern Levant specialization in olive and grape cultivation peaks in this phase and was likely largely oriented towards maritime trade. The Neo-Assyrians appear to have had a stronger direct interest in wine, as indicated in texts, depictions, and the expansion of settlements with grape evidence to the East, possibly related with expanded trade and contacts. A strong focus on grape cultivation is also visible in the Northern Levant.

While during much of the period under consideration cultivation of olive and grape was the focus of the Levant, related to ideal environmental conditions for olive and grape growth, periodic eastward expansions might signal the fruits' perceived importance and demonstrate the cultural integration of Southwest Asia.

Our research demonstrates the importance of olive and grape cultivation over the course of civilization of Southwest Asia. However, the history of olive and grape cultivation was also shaped by climatic conditions to variable degree. The close match between trends of population density and olive and grape evidence until 1300 BC for the Levant underlines that these fruits were essential to the population and played a central role in the economic prosperity of the region through trade networks. Trade, at the same time, contributed to and resulted from greater cultural integration of the region, enabling regional specialization in olive and grape production in the Southern Levant, which ultimately culminates in the period 900–600 BC related to the expansion of trade networks associated with the Neo-Assyrian territorial empire. Our study shows how olive and grape sustained and changed societies, from enabling

economic prosperity, nurturing cultural integration, and instigating specialization, all of which would go on to define human civilization in Southwest Asia.

## Supporting information

**S1 Table. Data for all the archaeobotanically represented sites, their location, elevation, chronology, present day rainfall as well as reconstructed rainfall as in (50) and their references.** Percentages of olive/grape seeds among the crops and ubiquities among the samples, as well as charcoal percentages and total number of charcoals are given for each site. (DOCX)

## Author Contributions

**Conceptualization:** Katleen Deckers.

**Data curation:** Katleen Deckers, Simone Riehl, Joseph Meadows, Valentina Tumolo.

**Formal analysis:** Katleen Deckers, Simone Riehl.

**Funding acquisition:** Dan Lawrence.

**Investigation:** Katleen Deckers, Simone Riehl.

**Methodology:** Katleen Deckers, Simone Riehl.

**Project administration:** Dan Lawrence.

**Supervision:** Dan Lawrence.

**Validation:** Katleen Deckers, Simone Riehl, Valentina Tumolo.

**Visualization:** Katleen Deckers, Joseph Meadows, Israel Hinojosa-Baliño.

**Writing – original draft:** Katleen Deckers.

**Writing – review & editing:** Katleen Deckers, Simone Riehl, Joseph Meadows, Valentina Tumolo, Israel Hinojosa-Baliño, Dan Lawrence.

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
