## [Decision Letter · Decision Letter 0]

4 Apr 2024

PONE-D-24-01610A history of olive and grape cultivation in Southwest Asia using charcoal and seed remainsPLOS ONE

Dear Dr. Deckers,

Thank you for submitting your manuscript to PLOS ONE. After careful consideration, we feel that it has merit but does not fully meet PLOS ONE’s publication criteria as it currently stands. Therefore, we invite you to submit a revised version of the manuscript that addresses the points raised during the review process.

**Reviewers 1 and 2 are quite positive about your work and provide suggestions and comments that will help with the clarity and transparency of your presentation. Please address all of these in your revisions. Reviewer 3 is more critical. The one issue that you need to address relative to their comments is that regarding Table 2. The table will be much clearer if you provide additional explanation of what you are demonstrating with the statistical tests. You can probably accomplish this in one or two sentences.**

We look forward to receiving your revised manuscript.

Kind regards,

John P. Hart, Ph.D.

Academic Editor

PLOS ONE

Journal Requirements:

Reviewers' comments:

Reviewer's Responses to Questions

**Comments to the Author**

1. Is the manuscript technically sound, and do the data support the conclusions?

Reviewer #1: Yes

Reviewer #2: Yes

Reviewer #3: Partly

2. Has the statistical analysis been performed appropriately and rigorously? 

Reviewer #1: Yes

Reviewer #2: Yes

Reviewer #3: Yes

3. Have the authors made all data underlying the findings in their manuscript fully available?

Reviewer #1: Yes

Reviewer #2: Yes

Reviewer #3: Yes

4. Is the manuscript presented in an intelligible fashion and written in standard English?

Reviewer #1: Yes

Reviewer #2: Yes

Reviewer #3: Yes

5. Review Comments to the Author

Reviewer #1: The paper represents a remarkable work due to the volume of processed data and its geographical and chronological extension, with interesting conclusions spanning environmental, climatic, social and cultural dynamics. The authors have meticulously cited a complete and equitable array of references, thereby contributing to an exhaustive overview of the current state of the art on this topic. This comprehensive review covers an area and period crucial for understanding the history of two of the most important fruit crops in the Mediterranean basin and surrounding regions. It is of great interest and certainly deserving of publication after very minor revisions.

Although some weaknesses are present, the authors have addressed them reasonably well, considering the heterogeneity of the proxies used. For instance, the low quality of some previously edited data obtained through non-rigorous field sampling methods was acknowledged, as well as the limitations in the resolution of some paleoclimatic reconstructions and the not always strong statistical robustness of certain results.

However, two issues require greater attention in my opinion. First, regarding seed counts, the authors mention using data from the ADEMNES database and some unpublished data reported in SI Tab. 1. However, it remains unclear whether their counts include solely intact seeds or also encompass seed fragments. This distinction significantly impacts data interpretation, especially when quantifying and comparing seed densities across different sites and periods. Therefore, the authors should explicitly state their approach and explain their choice.

Additionally, in lines 491-495, the authors assert that ‘Although phytogeographic changes happened throughout the Late Pleistocene and Holocene, archaeobotanical findings of potentially cultivated fruit seeds outside their natural distribution ranges can be considered a clearer indication of cultivation, especially when their wood charcoal is also present assuming wood was likely locally collected.’ however, later (line 523), they report that archaeobotanical evidence show a larger natural distribution than Zohary and Hopf. While there is no strong contradiction in this reconstruction, I recommend exercising caution when stating that evidence recorded outside the current distribution range of wild olive and grapevine is a clear indicator of cultivation. Our lack of knowledge about ancient distribution areas warrants moderation of this assumption, and it should be clearly reiterated at the beginning of the discussions.

Finally, it is crucial to conduct a thorough check on image references, as I’ve noticed some discrepancies that need verification (line 515). Therefore, I recommend verifying all of them.

Once the authors address these issues, the resulting body of knowledge will be more robust and transparent, undoubtedly contributing to further advancements in the field.

Reviewer #2: Overall, this is an outstanding article based on careful research and I definitely think it is suitable for publication in PLOSONE. The study of Deckers et al. is based on the creation of a vast database (ADEMNES) that includes all relevant findings. Seed and charcoal remains from the Levant, Mesopotamia, and Iran were integrated into this extensive database in order to investigate the production and consumption of olive and grape. Palynological studies were also taken into account.

The paper makes a significant contribution to our understanding of two (out of the five) founders of fruit tree horticulture, the olive (Olea) and the grape (Vitis). The study addresses important questions regarding when and where (mainly by integrating archaeobotanical findings with a model of the present-day natural distribution of wild olives and grapes) these two founders were first brought under domestication. Their history of cultivation during the last several millennia is also explored in detail. Despite the complexity of the subject matter, Deckers et al. present their findings in a clear and engaging scientific manner, making them accessible to archaeologists, archaeobotanists, and paleoclimatologists. All in all, the paper is well written, includes critical statistical analyses and discussions, and is based on vast and relevant literature (the most up-to-date to the best of my knowledge).

Though the paper of Deckers et al. is a testament to the power of an interdisciplinary approach, I do think there are a few minor issues that should be added/corrected in order to improve it. These include:

Lines 60-62: The Assyrian Empire, in general, was interested in a specialized economy. In some regions, it was devoted to oleo-culture, while others were exploited for viticulture (e.g., Finkelstein et al. 2022, PEQ).

Line 89: correct the brackets.

Line 223: Please recheck the date you mention for Tell esh-Shuna; as far as I know, it is about a millennium younger.

Results: Though the inclusion of illustrative figures, tables, and maps further enhance the clarity of the exposition, facilitating a deeper understanding of the archaeobotanical database, I felt that it was too much to follow. I suggest, therefore, that some of the figures (for example, Figs. 15, 16, 17, and 19) be removed from the Supporting Information.

Reviewer #3: There are significant changes that need to be made to Table 2; I suggest restructuring the information in terms that are more understandable.

While the chi-square is an interesting analysis, the original data used should be presented in tabular form to facilitate comparison.

Possibly, a Bayesian approach to the problem would be much more suitable and informative.

6. PLOS authors have the option to publish the peer review history of their article (what does this mean?). If published, this will include your full peer review and any attached files.

Reviewer #1: No

Reviewer #2: No

Reviewer #3: **Yes: **Javier Valera

---

## [Author Response · Author response to Decision Letter 0]

19 Apr 2024

We sincerely appreciate the reviewers for their thorough evaluation of our manuscript and for providing valuable suggestions to improve it. We now worked through the reviewer's comments and improved our manuscript, in most cases according to the reviewer's suggestions.

Reviewer #1: The paper represents a remarkable work due to the volume of processed data and its geographical and chronological extension, with interesting conclusions spanning environmental, climatic, social and cultural dynamics. The authors have meticulously cited a complete and equitable array of references, thereby contributing to an exhaustive overview of the current state of the art on this topic. This comprehensive review covers an area and period crucial for understanding the history of two of the most important fruit crops in the Mediterranean basin and surrounding regions. It is of great interest and certainly deserving of publication after very minor revisions.

Although some weaknesses are present, the authors have addressed them reasonably well, considering the heterogeneity of the proxies used. For instance, the low quality of some previously edited data obtained through non-rigorous field sampling methods was acknowledged, as well as the limitations in the resolution of some paleoclimatic reconstructions and the not always strong statistical robustness of certain results.

1. However, two issues require greater attention in my opinion. First, regarding seed counts, the authors mention using data from the ADEMNES database and some unpublished data reported in SI Tab. 1. However, it remains unclear whether their counts include solely intact seeds or also encompass seed fragments. This distinction significantly impacts data interpretation, especially when quantifying and comparing seed densities across different sites and periods. Therefore, the authors should explicitly state their approach and explain their choice.

The ADEMNES database generally includes all data as published. Since the late 1980s, standard counting of fragments has mostly been applied, e.g., adding two halves or four-quarters of a seed in one count. Hence, most of the studies included in this manuscript apply this. However, there are some objects, such as pistachio shell fragments, where there is less standardization: Some researchers for example weigh the fragments and translate the weight into whole nuts. There were also some publications included in this manuscript that predate the emergence of the standards. Not all researchers have published their counting methods, and we can never know if the table contains fragments or whole seeds. 

Hence, the particular case of possibly fragmented seeds adds to many other quantification problems in the field of archaeo-/palaeobotany, such as different numbers of reproductive units per plant or different transport distances, which means that seed or pollen counts can hardly be extrapolated to whole plants. Nevertheless, the whole discipline works with these counts because it is the only way to extract some meaning from the recording of the objects. We added a few sentences to the text that remind the readers to this issue:

“Data was entered into ADEMNES as published”. “Since the late 1980s, most publications have used standardized methods for counting fragmented seeds. This involves adding two halves or four quarters of a seed as one count for most seeds. However, before this period, various counting methods were used, and some variable practices are still in use today for the counting of pistachio shells. These variable practices may introduce some inaccuracies to the quantification but still provide approximate insights.”

2. Additionally, in lines 491-495, the authors assert that ‘Although phytogeographic changes happened throughout the Late Pleistocene and Holocene, archaeobotanical findings of potentially cultivated fruit seeds outside their natural distribution ranges can be considered a clearer indication of cultivation, especially when their wood charcoal is also present assuming wood was likely locally collected.’ however, later (line 523), they report that archaeobotanical evidence show a larger natural distribution than Zohary and Hopf. While there is no strong contradiction in this reconstruction, I recommend exercising caution when stating that evidence recorded outside the current distribution range of wild olive and grapevine is a clear indicator of cultivation. Our lack of knowledge about ancient distribution areas warrants moderation of this assumption, and it should be clearly reiterated at the beginning of the discussions.

We slightly changed the wording throughout the manuscript regarding this issue – mostly by replacing “evidence” to “indication”.

3. Finally, it is crucial to conduct a thorough check on image references, as I’ve noticed some discrepancies that need verification (line 515). Therefore, I recommend verifying all of them.

Thanks for noticing that. We checked through all figure references and found a few erroneous ones in the last part of the manuscript, which we now corrected.

Once the authors address these issues, the resulting body of knowledge will be more robust and transparent, undoubtedly contributing to further advancements in the field.

Reviewer #2: Overall, this is an outstanding article based on careful research and I definitely think it is suitable for publication in PLOSONE. The study of Deckers et al. is based on the creation of a vast database (ADEMNES) that includes all relevant findings. Seed and charcoal remains from the Levant, Mesopotamia, and Iran were integrated into this extensive database in order to investigate the production and consumption of olive and grape. Palynological studies were also taken into account.

The paper makes a significant contribution to our understanding of two (out of the five) founders of fruit tree horticulture, the olive (Olea) and the grape (Vitis). The study addresses important questions regarding when and where (mainly by integrating archaeobotanical findings with a model of the present-day natural distribution of wild olives and grapes) these two founders were first brought under domestication. Their history of cultivation during the last several millennia is also explored in detail. Despite the complexity of the subject matter, Deckers et al. present their findings in a clear and engaging scientific manner, making them accessible to archaeologists, archaeobotanists, and paleoclimatologists. All in all, the paper is well written, includes critical statistical analyses and discussions, and is based on vast and relevant literature (the most up-to-date to the best of my knowledge).

Though the paper of Deckers et al. is a testament to the power of an interdisciplinary approach, I do think there are a few minor issues that should be added/corrected in order to improve it. These include:

1) Lines 60-62: The Assyrian Empire, in general, was interested in a specialized economy. In some regions, it was devoted to oleo-culture, while others were exploited for viticulture (e.g., Finkelstein et al. 2022, PEQ). 

Thanks for this indication: We added this to the introduction and came back to it in the discussion:

“For Judah however, regional differences were seen in high-gain high-risk specializations under the Assyrian rule, depending on the regional conditions, e.g., with the Shephelah specializing in olive, the highlands in viticulture, the Dead Sea valley on dates and exotic plants, while the Beersheba Valley on trade (21)”

This is related to southern Levantine specialization in olive and grape cultivation, also regionally diverse within that region, cf. also (21). This regional specialization was likely also a consequence of the expanded trade networks associated with the Neo-Assyrian territorial empire (compare with e.g., (115)) and increased specialization in trade of particular groups (21), hence a result of greater cultural integration.”

2) Line 89: correct the brackets. We altered it as suggested.

3) Line 223: Please recheck the date you mention for Tell esh-Shuna; as far as I know, it is about a millennium younger.

Thanks for making sure we apply the correct chronology. We checked the chronology: There are 2 occupation phases at Tell esh-Shuna, one that dates to the Chalcolithic period and another that dates to the Early Bronze Age. Here is a reference that refers to radiocarbon dates published that demonstrates this: Bronk Ramsey, C.; Higham, T.F.G., Owen, D.C., Pike, W.G., Hedges, R.E.M (2002) Radiocarbon Dates from the Oxford Ams System: Archaeometry Datelist 31. Archaeometry 44, 83-84. 

We however improved our originally used Chalcolithic chronology for Tell esh-Shuna throughout the manuscript including the figures and tables: Instead of applying the period date 5000-4500 BC with average 4750 BC, now we applied the 5100-4750 BC phase date, with an average date of 4925 BC. The latter represents the more precise chronology of the occupation. This has no impact on our conclusions.

4) Results: Though the inclusion of illustrative figures, tables, and maps further enhance the clarity of the exposition, facilitating a deeper understanding of the archaeobotanical database, I felt that it was too much to follow. I suggest, therefore, that some of the figures (for example, Figs. 15, 16, 17, and 19) be removed from the Supporting Information.

We are of the opinion that these figures should ideally remain within the main text, as they each serve to clarify important aspects:

* Fig 15 shows clearly the elevational differences (i.e. the move towards lower elevations) that we discuss in the text. This is an easy-to-read figure for an important point.

*Fig. 16 shows more details but makes it easier to understand differences in the scale of involvement for the different elevations and rainfall (not just the presence/absence). Scale and rainfall are important factors that are referred to in the text.

*Fig. 17 shows the economical specialization over time. This is an important point in our manuscript.

*Fig. 19 shows that some palaces are involved in production, and it compares palace values against all values. 

We think it is unlikely that many readers will delve into supplementary materials to locate supplementary figures. However, if the editor deems it necessary to relocate these figures, we would do so.

Reviewer #3: There are significant changes that need to be made to Table 2; I suggest restructuring the information in terms that are more understandable.

1. While the chi-square is an interesting analysis, the original data used should be presented in tabular form to facilitate comparison.

We reorganized the table so that it is hopefully easier to read and understand. We also included the mean score difference, standard error difference, Z- and p-value. All raw data used for the statistical analysis is presented in the S1 Table. We just combined the column (Region and Period BC). We added this information to the caption. All statistical analysis referred to in the manuscript are reproducible starting from this table.

2. Possibly, a Bayesian approach to the problem would be much more suitable and informative.

This comment is somewhat vague, and it's not entirely clear to us how Bayesian approaches would specifically enhance our current work. We believe it would be best to perhaps address this suggestion in future iterations of our research.

---

## [Editor Report · Decision Letter 1]

29 Apr 2024

A history of olive and grape cultivation in Southwest Asia using charcoal and seed remains

PONE-D-24-01610R1

Dear Dr. Deckers,

We’re pleased to inform you that your manuscript has been judged scientifically suitable for publication and will be formally accepted for publication once it meets all outstanding technical requirements.

Kind regards,

John P. Hart, Ph.D.

Academic Editor

PLOS ONE